# What can be learned from fishers' perceptions for fishery management planning? Case study insights from Sainte-Marie, Madagascar

Thaïs A. Bernos[1¤]*, Clodio Travouck[1], Naly Ramasinoro[1], Dylan J. Fraser[2], Barbara Mathevon[3]

1 Natural Resources Management Program, Gret Professionals for Fair Development, Soavimbahoaka, Madagascar, 2 Department of Biology, Concordia University, Montreal, Quebec, Canada, 3 Natural Resources Management Program, Gret Professionals for Fair Development, Nogent sur Marne Cedex, France

¤ Current address: Department of Ecology and Evolutionary Biology, University of Toronto, Toronto, Ontario, Canada
* thais@bernos.fr

**Data Availability Statement:** Data will be held in a public repository.

**Funding:** Funding to T.B. was provided by Agence France Volontaire, Noé Conservation, Find-Gret,

## Abstract

Local support is critical to the success and longevity of fishery management initiatives. Previous research suggests that how resource users perceive ecological changes, explain them, and cope with them, influences local support. The objectives of this study were twofold. First, we collated local fishers' knowledge to characterize the long-term socio-ecological dynamics of the small-scale fishery of Sainte-Marie Island, in Madagascar. Second, we empirically assessed the individual- and site-level factors influencing support for fishery restrictions. Our results indicate that fishers observed a decline in fish abundance and catch sizes, especially in nearshore areas; many also perceived a reduction in fish sizes and the local disappearance of species. To maintain their catches, most fishers adapted by fishing harder and further offshore. Accordingly, fishers identified increased fishing effort (number of fishers and gear evolution) as the main cause of fishery changes. Collectively, our results highlight that the transition from a subsistence to commercial fishery, and resulting changes in the relationship between people and the fisheries, was an underlying driver of fishery changes. Additionally, we found that gender, membership to local associations, coping mechanisms, and perceptions of ecological health, were all interlinked and significantly associated with conservation-oriented attitudes. Conservation-oriented attitudes, however, were not associated with fishers' willingness to decrease fishing. In the short-term, area-based restrictions could contribute to building support for conservation. In the long-term, addressing the underlying causes of the decline will necessitate collaborations among the various groups involved to progressively build livelihood flexibility. Collectively, our study provides additional insights on the individual- and site-level factors influencing support for fishery restrictions. It also highlights the importance of dialoguing with fishers to ensure that fishery management plans are adapted to the local context.

and Cétamada. The funders had no role in study design, data collection and analysis, decision to publish, or preparation of the manuscript.

## Introduction

Many coastal communities depend on marine resources for their livelihood and well-being. Ensuring that local needs for marine resources are met without compromising those of future generations often requires the development of fishery management plans–sets of rules guiding the conservation and sustainable use of marine resources. Lack of local support can result in non-compliance with rules, political struggles, and increased social tension [1, 2], thereby threatening social and ecological outcomes and the long-term persistence of fishery management interventions. Thus, understanding the factors shaping local support for management is crucial to plan and implement effective fishery management interventions [3, 4].

Within the fishery context, research has shown that local support for management is influenced by both individual- and site-level factors. Key individual factors influencing support for conservation include the level of attachment to fishing, resource dependency, and employability [5–8]. Perceptions of changes are also important: individuals are more likely to be supportive if they perceive ecosystem health as impaired, interpret local human behaviours as a driver of ecological degradation, and perceive management actions as beneficial to ecosystem health [9–12]. Site-level factors influencing support for conservation include local assets (e.g. physical, social, natural, political), underlying values, cultural practices, and ecosystem health [13–15]. For instance, fishers might be more reliant on marine resources and in economically poorer areas, with fewer livelihood alternatives [6, 7, 16–18].

Madagascar's shorelines harbour areas of mangroves, coral reefs, sea grass, and a rich marine biodiversity, that are critically important to both feed and support the livelihood of coastal communities [19]. However, the ecological health of the area is threatened by overexploitation, destructive fishing practices, sediment supplies, climate change, and declining fisheries [20–25]. Substantial efforts have been made to manage marine resources more sustainably, including the creation of a network of managed coastal water areas referred to as Locally Managed Marine Areas (LMMAs) [26–28]. Collectively, this LMMA network currently covers an estimated 17% of Madagascar's shorelines [29]. While the specific regulations vary with the social, ecological, and economic objectives of each LMMA, marine resources are always effectively managed by local resource users [30–32]. As Madagascar coastal areas are densely populated and highly reliant on fisheries, with limited state-capacity to enforce fishery legislation, fishery rules are expected to be largely self- or community-enforced.

The spatial and temporal knowledge of those with a long familiarity with the local ecology is well-recognized as an important component of fishery management planning [33–35]. In data-less management scenarios, including many nearshore tropical areas, fishers' knowledge can reveal important information about fishery resources and the history of changes in the socio-ecological system [36–38]. Previous studies of fishers' perceptions in Madagascar examined areas with longstanding experiences with fishery management initiatives [25, 39]; in contrast, this study was conducted at an early planning stage of a fishery management initiative for the small-scale fishery of Sainte-Marie (*Nosy boraha*), an island located off northeastern Madagascar. When this study was conducted, the historical dynamics of the fisheries remained uncharacterized.

In this study, our objectives were two-fold. First, we characterized the long-term socio-ecological dynamics of Sainte-Marie's fishery. Second, we empirically assessed what factors might influence support for fishery restrictions. To do so, we 1) assessed perceptions related to changes in the fisheries, their causes, potential solutions, and fishers' coping mechanisms and 2) examined whether fishers' propensity to suggest management restrictions correlated with key factors known to influence support for conservation. We focused on four groups of variables shown to influence such conservation-oriented attitudes or behaviours; individuals'

characteristics (age, dependence on the fisheries, gender, membership to local associations, and attachment to the activity), coping mechanisms (adaptation to past and future changes), perceptions related to ecological health (fish number and size, species disappearance, changes in fishing grounds), and site-level characteristics (number of hotels, shops, nightlife, and presence of lagoons) [37, 40, 41]. By doing so, this work contributes to the understanding of the synergies among factors influencing support for fishery management with significant implications for the effectiveness of management initiatives. Further, our study highlights the importance of local knowledge to develop a shared understanding of the socio-ecological system and address management challenges.

## Study area and context

Sainte-Marie Island harbours 17 town (16 of which located directly on the shorelines) arranged in four boroughs (Fig 1). The towns differ in site-level characteristics that might influence perceptions related to fisheries and individual's ability to cope with changes. Notably, infrastructures (e.g., large hotels, airport, port, seafood processing factory) are concentrated in the main town, Ambodifotatra (latitude, longitude = -16.998462˚, 49.852690˚), and the southern boroughs, providing more access to alternate livelihood options. While all boroughs experience a climate favorable to cash crops and trees (i.e. clove trees, vanilla), their exposure to seasonal extreme weather events (heavy rainfalls, annual cyclones) and bad weather varies. In particular, the western boroughs, separated from Madagascar's mainland by the 7-23km wide Sainte-Marie canal, are more sheltered than the eastern boroughs. The southeastern shoreline harbours most of the near-shore coral reefs and lagoons, which attract tourists and influence local fishing practices.

Fishing practices and targets are extremely diversified. Most fishermen use non-motorized pirogues, and the main fishing techniques are line-fishing, netting (gill nets, beach seine), free diving, and fish traps. Fishers typically use several fishing techniques: technique use is influenced by fishers' conditions, preferences, target species, the weather, and fishing locations. For instance, older fishers are more likely to use fish traps while many younger fishermen dive; under bad weather conditions, fishers are more likely to be found netting in the lagoons that line fishing offshore. Fishers generally harvest multiple species. They used more than 50 common names referring to the most commonly caught fish and marine resources (see S1 Table for the species caught the most frequently), which included both mobile (e.g. migratory fishes) and sedentary species (e.g. mollusks, arthropods). As for market, fishers mainly sell their catches to a commercial fishery with an international export market, hotels, restaurants, and locals. In most towns, local collectors working for the exporting fishery are given fridges to store marine resources, which are then collected, processed, and packaged for export. As many other small-scale fisheries, fishery participation is gendered [44–46]; fisherwomen tend to fish for subsistence, target fish species of lower economic value as well as invertebrates, and search for marine resources in easily accessible near-shore areas.

Sainte-Marie's nearshore fishery is managed under national fishery laws, which are not fully complied with and enforced as responsible authorities lack the capacity to enforce them. It is worth noting that a protected area was established in 2014 by the Indian Ocean Commission and local stakeholders. It generated significant hostility from the fishers it impacted in the Southern borough. Eventually, the resistance of the communities led to conflicts and the resulting cancellation of the initiative. Other than western approaches to management, local cultural taboos (*fady*) may also play a role in regulating fishery-related activities. Some *fady* entail species-specific protection: killing Guitarfish is taboo for all, and eating sea turtles for certain families. Other taboos are location specific. For instance, on the sandy islets off the

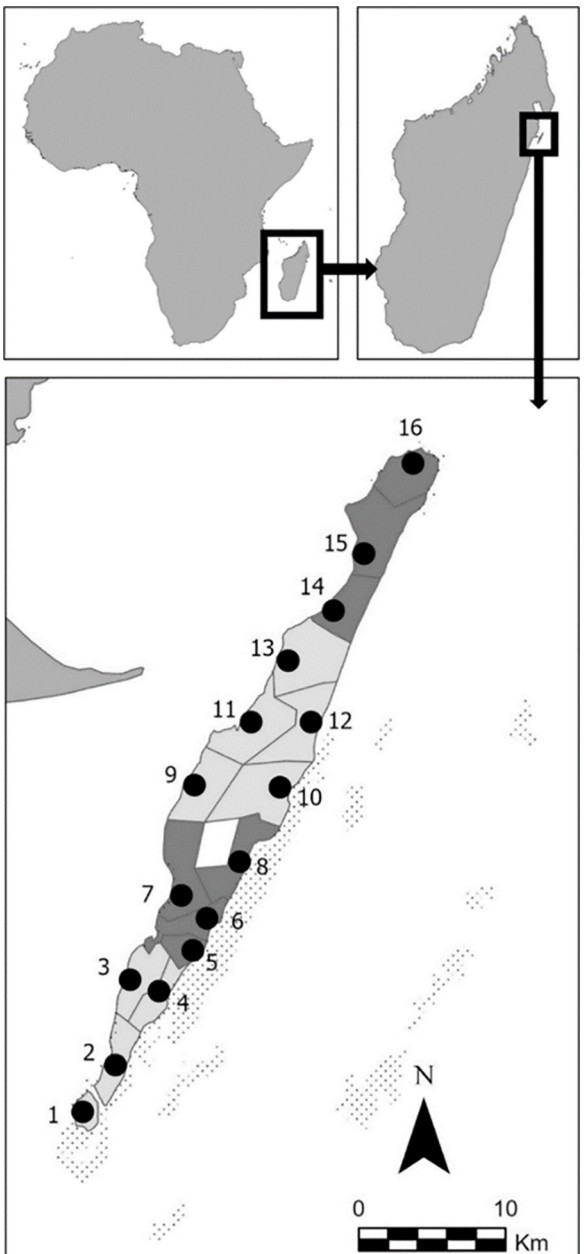

**Fig 1. Map of Sainte-Marie Island, Madagascar.** The boundaries of the 17 towns (contour line), the four boroughs (shades of grey), as well as coastal lagoons (dotted), are shown. Black dots indicate the town sampled as follow; 1 = Agniribe, 2 = Vohilava, 3 = Mahavelo, 4 = Ambodiforaha (borough = Vohilava); 5 = St Joseph, 6 = Ankobaoba, 7 = Ambodifotatra, 8 = Agnalaradjy (borough = Ambodifotatra); 9 = Maromandia, 10 = Agnafiafy, 11 = Loukintsy, 12 = Sahasifotatra, 13 = Agnivorano (borough = Loukintsy); 14 = Ifotatra, 15 = Ambatourao, 16 = Ambodiatafana (borough = Ambatourao). Data layers: Coral reefs [42], Madagascar subnational administrative boundaries [43].

southeastern end of the island, harvesting several types of marine resources at once, fishing with nets, and polluting the area, are prohibited. Access is also restricted (prohibited to Merina, a highlander ethnic group). When taboos are violated, spiritual sanctions involve mischance, disease, or death; material sanction can also be ordered by the community. However, fishers express that taboo enforcement and respect has declined because of several external

changes, including the apparition of a new market for guitarfish fins, market multiplication, and harder times. In conclusion, marine resources are essentially open access.

In 2015, the non-governmental organization GRET -Professionels du développement soli-daire (professionals for fair development)- started working with communities to improve liveli-hood sustainability. This led to the creation of the PCADDISM -Plateforme de concertation des acteurs pour le développement durable de l'île Sainte-Marie (Concertation platform for sustain-able development actors on Sainte Marie Island)-, a platform managed and led by communities where local stakeholders (e.g. local fishers' federation, commercial fishery, private hotels) are represented. The GRET provides technical and financial support to the PCADISM. When the present research was conducted, the idea to create LMMAs governed by traditional laws (*dina*) had started to gain momentum within the PCADISM. The PCADISM was holding meetings in each town to discuss livelihood sustainability, including possible fishery management strategies. Those meetings were open to all who wished to come, highly deliberative, and widely attended.

## Material and methods

### Survey strategy

Between June and October 2017, we interviewed fishers in 16 of the 17 towns (Fig 1). We could not find any fishers in Ambohitry, the only town not located directly on the shorelines. As we found that fishers' availability was weather-dependent (they usually were at sea when the weather was good), we conducted data collection during the raining season. Each town was visited at least three times to sample as many fishers as possible. Upon arrival in a town, we introduced ourselves to local authorities and explained the purpose and intended use of the survey. After obtaining their permission to conduct research in the area, we asked them to sug-gest areas where fishers were likely to be located and to identify suitable survey respondents. We asked interviewees to refer us to additional fishers (snowball sampling). Sampling was pur-posive rather than random as we wanted to ensure sufficient representation of various fishing practices and socioeconomic backgrounds. As part of this effort, we employed a gender-inclu-sive approach [45]. For instance, when looking for suitable respondents, we defined fishers broadly as people who extract marine resources using various fishing methods for commercial or subsistence purposes. We also stated that we were looking for both women and men. The interviews were semi-structured and conducted in Malagasy. After being told about the pur-pose of the survey and its intended use, interviewees were asked whether they consented to participate. Interviews lasted between 30min and 3 hours.

### Interview structure

Our initial questions (S2 Table) focused on fishers' socioeconomic characteristics and fishing history, for example, when they began fishing, what species they targeted, whether they used fishing vessels, and with what gear. Our next questions focused on fisher perceptions of changes, including whether they had noticed changes in fish abundance, fish sizes, or knew of local species extinction. We also asked whether they had altered their fishing effort and fishing locations.

The next part of the interview focused upon temporal trends in catch sizes and fishing ground location. Both empirical [47, 48] and cognitive studies [49, 50] showed that people typ-ically have trouble recalling events and their timing unless provided with appropriate cues. Hence, we asked fishers to recall catch sizes for their primary target species (for up to three tar-gets) during the past year [51]; we then asked fishers to recall their catches for the period when they first began, and around 2009 when applicable. We used 2009 as a cue because that year an intense cyclone caused tremendous damages on the island, thereby impacting the memory of

its inhabitants. We focused on great catch sizes as we found that fishers had trouble estimating and recalling average catch sizes; furthermore, studies have shown that people tend to recall great or poor catch sizes more accurately that typical catch sizes [52]. We also asked fishers about their fishing site location (distance from the shore) during the past year, and for the period when they first began fishing.

We then asked fishers what, in their opinion, could be the underlying causes of the changes and what could be done to improve ecological health in the area. Finally, we determined fishers' coping mechanisms to future changes by asking what they would do in response to a hypothetical scenario of further decline.

## Ethical considerations

Data collection aimed to inform the type of interventions that a non-governmental association, the GRET, could deliver to help reduce poverties and inequalities locally. The GRET is service-oriented; it does not have an institutional ethics committee, and there is no human research ethics committee overseeing the activities of non-governmental associations in Madagascar. However, we followed published recommendations for ethical research conduct in Madagascar [53]. All the activities carried out by the GRET at Sainte-Marie (including this study) were reviewed and approved by the decentralized technical services of the State and local authorities prior to being conducted. We shared the results of our study with local stakeholders and at higher levels, and adhered to common principles of human research ethics [54, 55]. For instance, we obtained verbal consent from the research participants prior to conducting the survey, ensured that the surveyed population mirrored the population targeted by the initiative, and obtained verbal agreement from the community leaders to conduct research. The data presented in this study is anonymous and aggregated to prevent direct linkage with individuals.

## Statistical analysis

We used generalized additive models (GAMs) to model temporal changes in best catch sizes for 13 of the most frequent target species family (local Malagasy name) identified from the interviews (S1 Table). GAMs were ideally suited to the structure of our data and the nature of our analysis because; 1) their non-parametric smoothing function (hereafter referred to as smoothers) allowed us to model nonlinear temporal trends [56, 57]; 2) they can incorporate both continuous and categorical variables; 3) they can accommodate random effects; and, 4) they estimate the shape of the relationship from the data itself (we did not have to specify any a-priori shape). For these reasons, GAM represented a flexible and powerful approach to model temporal trends in best catches, as well as their nature and timing. We modeled temporal trends with negative binomial GAMs and a logit link using the mgcv package [56] (S1 File for R script). A first model (Mod1) included different intercept for each species, a smoother for time, and its interaction with species. The species term corresponded to the common Malagasy name, which often applied to multiple fish species (S1 Table). As perceptions varied among fishermen, we fitted fisherman identity as a random intercept and slope. To investigate plausible alternative hypotheses, we constructed several additional candidate models. Mod2 included the species intercept and a smoother for time; Mod3 and Mod4 included either one of the species intercept or time smoother. We selected the best model based on the lowest AIC criterion, highest restricted log-likelihood, and highest explanatory power. Furthermore, we evaluated the significance of the fixed effect using Wald's test.

To model temporal changes in fishing distance from the shore, we used generalized linear models with a Poisson distribution because fishing distances were positively skewed (S1 File for R script). As the magnitude of the perceived changes could vary by gear, we fitted a first

model (Mod1b) with an interaction between fishing gear and time as a fixed effect. Mod2b included an intercept for gear and time; Mod 3 and 4 included either one of the gear intercept or time. We selected the best model based on the lower AIC and evaluated the significance of the fixed effect based on likelihood ratio-tests.

We then used Multiple Factor Analysis (MFA) to explore how an individual's propensity to suggest management restrictions was associated with individual characteristics, site-level characteristics, coping mechanisms, perceptions related to ecological health and underlying causes. Tailored to accommodate qualitative variables, MFA is an extension of Principal Component Analysis in which variables within the same groups are weighted to balance the importance of the groups. It enabled us to account for the existing structure of our data, where we had several groups of variables; for instance, the group "perceptions of ecological health" included responses about local extinction, changes in size, fish numbers, and fishing site distance (Table 1). We performed the MFA using the "FactoMineR" package [58] in R [59] (S2 File for R script). We used Horn's parallel analysis implemented in the "paran" package [60] to determine the number of dimensions retained in the MFA. We included support for restrictions as a supplementary variable: individuals that are close to one another in multidimensional space share similar individual attributes, perceptions of ecological health, underlying causes, coping mechanisms, or local assets. The main dimensions of this variability are then related to support for restrictions, specified as a supplementary variable. We used the v.test statistic to identify significant association (v.test >2) between support for restrictions and the dimensions [61]. To understand dimensional grouping and associations between variables, we used the visual representation of the MFA and identified the variables that were the most significantly associated ($R^2 > 0.20$, $p < 0.05$) with each dimension. Prior to running the MFA, we ran the Keyser-Meyer-Olkin (KMO) and Bartlett's test of homogeneity of variances implemented in "psych" [62] to confirm sampling adequacy.

Prior to running the MFA, answers had to be classified (Table 1). Questions related to perceptions of ecological health, underlying causes, and solutions were classified as positive and

**Table 1. Groups of variables tested in the multiple factor analysis (MFA) and associated accronyms.**

| Group | Variables | Answers | Acronym |
|---|---|---|---|
| **Individual attributes** | Gender | Female/male | IN_woman/IN_man |
| | Age | Younger (<40)/Older (> = 40) | IN_young/IN_old |
| | Dependence | Yes/No | IN_dep/IN_nodep |
| | Gear ownership | Owns/does not own expensive gear | IN_own/IN_noown |
| | Association membership | Yes/No | IN_asso/IN_noasso |
| | Attachment to fishing | Yes/No | IN_att/IN_noatt |
| **Perceptions of ecological degradations** | Fishing site distance | Further/not further | ED_dist/ED_nodist |
| | Fish abundance | Fewer/not fewer | ED_num/ED_nonum |
| | Local extinction | Yes/No | ED_disp/ED_nodisp |
| | Fish size | Smaller/not smaller | ED_size/ED_nosize |
| **Perceptions of causes** | Causes | Linked/not linked to local fishing | CO_fish/CO_nofish |
| **Support for restrictions** | Solutions | Linked/not linked to fishery restrictions | RE_yes/RE_no |
| **Coping mechanisms** | Realized | Decrease fishing effort/continue/adapt | CR_decrease/CR_continue/CR_adapt |
| | Hypothetical | Decrease fishing effort/continue/adapt | CH_decrease/CH_continue/CH_adapt |
| **Local assets** | Hotels | More than/less or equal to three | LA_hot/LA_nohot |
| | Leisure | More than/less or equal to one | LA_leis/LA_noleis |
| | Shops | More than/less or equal to 24 | LA_sho/LA_nosho |
| | Lagoons | Lagoons (< 5km) / No Lagoons (> 5km) | LA_lag/LA_nolag |

labelled as such when they could support management. For instance, when asking about solutions to the observed decline, fishers who suggested management restrictions as a mean to improve ecological health were classified as having a conservation-oriented attitude and labelled as "yes", and those who did not suggest management restrictions were labelled as "no" (Table 1). Similarly, fishers who identified local fishing as a cause of the decline were labelled as "yes", and those who did not as a "no". For the coping mechanisms, we classified the responses to changes (past and future) as; 1) adapt in a way that could amplify a decline (i.e. fish harder, move location, change gear), 2) continue fishing as before, or 3) decrease fishing [63]. In Sainte-Marie, "adapt" is not a conservation-prone behaviour as it could amplify the scale of the decline by sequentially depleting marine resources [63, 64]. Similarly, "continue" could also amplify the decline; however, it is more of a coping mechanism than an adaptation. The site-level variables were codified based on whether they were under or above Sainte-Marie's median value for each town using data from a socioeconomic survey of Sainte-Marie (GRET, unpublished data 2021).

## Results

### Fishers' characteristics

In total, we interviewed 127 fishers, including 109 men and 18 women (S3 Table). Despite our sampling efforts to collect additional data on women (including additional trips to specifically seek out fisherwomen), surveyed men outnumbered women. This result partly reflects the male dominance in the fishery. Indeed, based on a recent socioeconomic survey, we estimate our sample sizes to represent 2 and 4% of, respectively, Sainte-Marie's fishermen and women (GRET, unpublished data, 2021).

As typical of many small-scale fisheries [45], fishing practices differed among surveyed fisherwomen and men (Table 2). Most men used more than one fishing gear; the predominant gear was line fishing (80%), followed by nets (50%), diving (28%), traps (16%), and spears (2%). Most women used nets (88%) followed by spears (12%). Men and women differed in fishing grounds; most men fished offshore from non-motorized pirogues (92%) while women generally fished on foot close to their towns (94%). These contrasting fishery practices are representative of gender-based differences practices in the area (GRET, unpublished data, 2021). The interview showed that men typically sold their catches to more than one market, of which the towns and the exporting fishery were the most frequently cited (82 and 60%, respectively); by contrast, women predominantly fished to feed their families. During data collection, we only encountered three migrant fishermen (two of which had been in Sainte-Marie for more than a decade).

Surveyed men and women differed in their level of formal education, reported income, and investment in the fisheries. The number of women with no formal education was proportionally greater (60 vs 72%, respectively men and women). Women, who reported lower average incomes (122 VS 61 USD$), were also less likely to have alternative income sources (51 and 22% of, respectively, men and women). Men were more committed to the fishery; they were more frequently members of local fisher's associations than women (64 vs. 6%) and owned expensive fishing gear (i.e boats, nets) more regularly (83 vs 39%). As differences in the circumstances and involvement of women emerging from this study and a socioeconomic survey of the area (GRET, unpublished data, 2021) could influence their perceptions, we discuss results separately among gender in the following section.

### Temporal changes in catch sizes and fishing site distance

We found evidence for markedly nonlinear trends in best catch sizes that cannot be interpreted easily without fishers' explanations of changes in the broader socio-ecological system

**Table 2. Fishing practices and socioeconomic characteristics by gender for Sainte-Marie's fishers.**

| Topic | Question | Response | Men (%) | Women (%) |
|---|---|---|---|---|
| **Practices** | Type | Motorized pirogues/boats | 7 | 0 |
| | | Non-motorized pirogues | 90 | 6 |
| | | Gleaning | 3 | 94 |
| | Gear used[×] | Line-fishing | 80 | 0 |
| | | Nets | 50 | 89 |
| | | Diving | 28 | 0 |
| | | Traps | 16 | 0 |
| | | Spears and sticks for octopus | 2 | 11 |
| | Gear ownership | Yes | 83 | 39 |
| | | No | 17 | 61 |
| | Markets[×] | Towns | 82 | 78 |
| | | International exporting fishery | 60 | 11 |
| | | Hotels/restaurants | 37 | 11 |
| | | Other collectors | 24 | 0 |
| **Characteristics** | Education | No formal education | 60 | 72 |
| | | Primary school | 30 | 28 |
| | | First cycle of secondary | 7 | 0 |
| | | Second cycle of secondary | 3 | 0 |
| | Dependence | Yes | 51 | 22 |
| | | No | 49 | 78 |
| | Income (USD/month) | | 122 | 61 |
| | Status | Local | 98 | 100 |
| | | Immigrant | 2 | 0 |
| | Membership to fishing associations | Yes | 65 | 6 |
| | | No | 35 | 94 |

[×] Questions for which multiple answers per person were possible (total may not add up to 100).

Results are based on interviews with 127 fishers, including 109 men and 18 women.

and their own fishing practices, which are summarized in the discussion (Fig 2). Mod2 outperformed the other models: it yielded the lowest AIC, the higher log-restricted likelihood, and the variance explained ($R^2$ and deviance explained) was nearly as high as Mod1 (S4 Table). The F tests further confirmed this conclusion: the interaction between time and species family was insignificant (Wald test p-value = 0.58), while the time smoother and species family intercepts were significant (Wald test p-values < 0.01). Generally, the smooth term for time showed an increase in best catch sizes from 1964 to the mid-1990s (Fig 2). The increase was most pronounced between 1980 and 1990; from the mid-1990s to 2017, best catch sizes decreased. This reconstruction of best catch sizes was based on 562 observations, from 101 fishers (mean = 5.6 observations per fisherman), for the 13 most frequently caught Malagasy term for fish or marine resources (mean = 43.23 observations per species group).

The assessment of fishing site locations indicates a progressive change from nearshore to offshore fishing for some of the fishing gear (Fig 3). Mod1b outperformed the other models based on AIC and variance explained (S5 Table). The interaction between fishing gear and

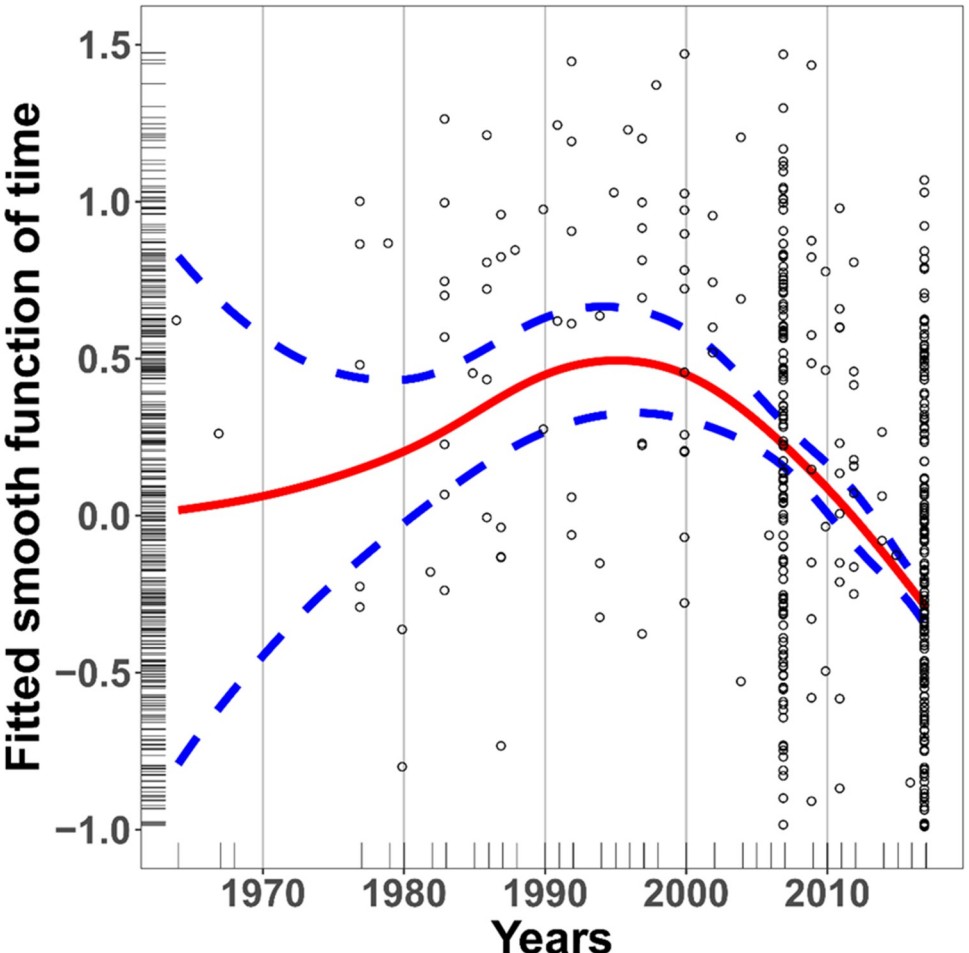

**Fig 2. Sainte-Marie's fishermen perceptions of catch changes over time estimated from generalized additive model (GAM).** The solid red line is the fitted smooth function for time, the blue dotted lines represent the 95% confidence intervals, the rugs on the margins display the location of individual observations, and the residual points are shown. The y limits of the plot are set (-1<y<1.5) to facilitate the visual interpretation of trends in best catch sizes.

time was significant (deviance = -6095, df = 199, p<0.001). Notably, we found that the magnitude of the changes perceived were greater for those fishing with hooks, followed by nets, and then free divers. Alternatively, fishers using fish traps were now fishing closer to the shore.

## Fishers' perceptions

**Fishery changes.** Overall, respondents painted the picture of a degraded fishery. Almost all fishers (93 and 100% of men and women) perceived a general reduction in fish abundance (Table 2). Men tended to indicate additional changes in the fishery (Table 2), which could reflect a different understanding of the fisheries. For instance, several fishermen indicated a decrease in fish size (58%) and many (55%) cited a total of 31 common names for species that had disappeared from the area (S6 Table). In particular, many fishers (>20) highlighted the disappearance of Dugongs (*Dugong dugong)* and two Mullidae species ("Enamahely" and "Antafan"). Many fishermen (53%) indicated that they had moved their fishing grounds offshore.

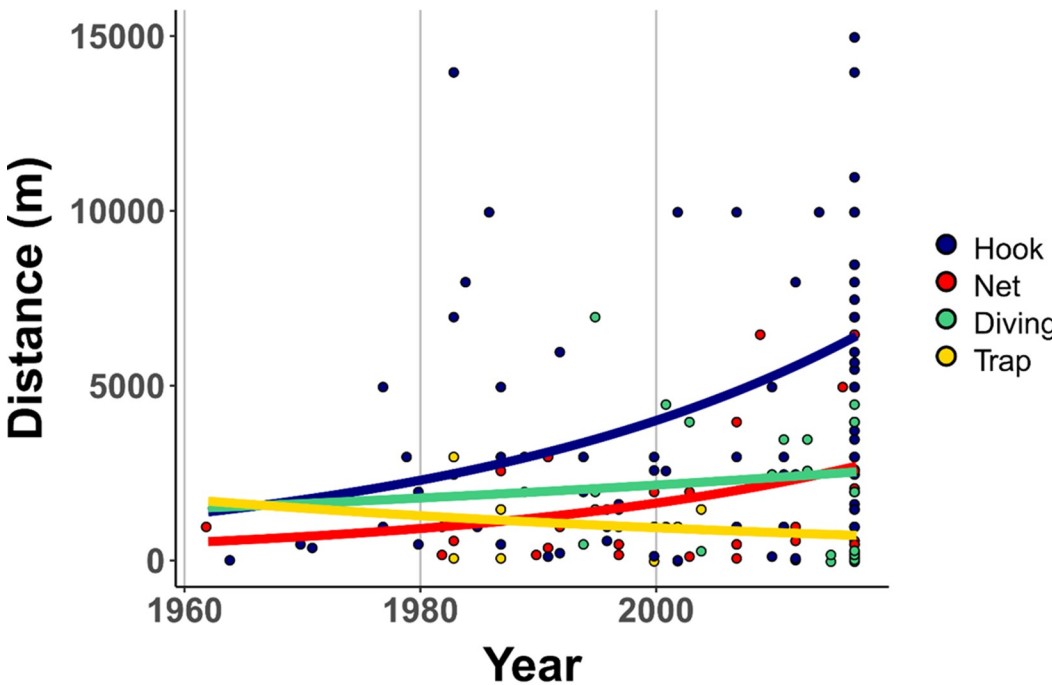

**Fig 3. Sainte-Marie's fishermen reported changes in fishing distance from the shore.** Regression slopes indicate differences the trajectories of changes in fishing distance for each of the four main fishing gear (hook, net, diving, and traps) inferred from generalized linear models. The y limits of the plot are set (0<y<15,000) to facilitate the visual interpretation of trends in fishing distance from the shore.

**Causes.**   When fishers were explicitly asked about the causes of the changes (Table 3), they predominantly attributed it to local fishing (84 vs. 59% of men and women). Specifically, fishers mentioned an increased number of fishers competing for marine resources (60 and 33%) followed by the evolution of fishing techniques. In particular, many felt that net-fishing practices had become less selective and more destructive (34% of men). Beside local fishing, several fishers (24 and 11%) attributed the decline to overlapping fisheries: they referred to the commercial exploitation of shrimp in the canal (8% of men) and sea-cucumbers on the nearshore areas (20 and 11%). They described the sea cucumber fishers as migrants who sporadically visit Sainte-Marie. There was a strong perception that those fishers competed with local fishers by using diving cylinders to collect marine resources and did not contribute to the local economy. Finally, fishers mentioned several more general causes to the changes: in particular, many mentioned the socioeconomic circumstances (i.e. lack of other alternatives) (28% and 6% of men and women) and spiritual causes (6% and 44%).

**Solutions.**   When assessing proposed activities to improve fishery health, men were overall more likely to suggest management restrictions than women (69 vs. 18% of men and women) (Table 3). Compared to men, women were more likely to offer no solutions (18 vs 33%) or propose to appease the gods (8 vs 28%), for instance via spiritual sacrifices and increased respect for local taboos. By contrast, many men suggested solutions that were related to local fishery restrictions (68 vs 17%); in particular, many proposed spatial restrictions (32%) and restrictions on gears, including nets (26%). Several fishers (12% and 22% of men and women) also suggested economic development to reduce the number of fishers.

**Coping mechanisms.**   There was a clear contrast between fishers' responses to past and future changes in the fishery, as well as among genders. Generally, men had adapted their

**Table 3. Observed changes to the fisheries, inferred causes, suggested solutions, and coping mechanisms by gender for Sainte-Marie's fishers.**

| Topic | Question | Response | Men (%) | Women (%) |
|---|---|---|---|---|
| **Changes** | Fish numbers | Decline | 93 | 100 |
| | | Increase | 2 | 0 |
| | | No changes | 6 | 0 |
| | Fish size | Reduction | 58 | 0 |
| | | Growth | 4 | 0 |
| | | No changes | 38 | 100 |
| | Fishing site distance | Further | 53 | 0 |
| | | Closer | 6 | 0 |
| | | No changes | 40 | 100 |
| | Extinct species | Yes | 55 | 0 |
| | | No | 44 | 100 |
| **Causes** [×] | Local human agency | More fishers | 60 | 33 |
| | | Net fishing | 34 | 39 |
| | | Gear diversification | 21 | 0 |
| | | Free-diving fishing | 18 | 11 |
| | | Destructive fishing | 6 | 0 |
| | | Offshore hooks | 0 | 6 |
| | Other human agency | Migrants | 20 | 11 |
| | | Prawn trawlers | 8 | 0 |
| | General | Socioeconomic | 28 | 6 |
| | | Spiritual | 16 | 44 |
| | | Weather unpredictability | 12 | 6 |
| | | Cyclones | 12 | 11 |
| | | Sea cucumber declines | 7 | 11 |
| | | Stochastic | 7 | 0 |
| | Don't know | | 0 | 6 |
| **Solutions** [×] | Local human agency | Spatial closures | 32 | 0 |
| | | Restrictions on nets | 26 | 17 |
| | | Restrictions on free-diving | 8 | 0 |
| | | Improved compliance | 9 | 0 |
| | Other human agency [×] | Restrictions for migrants | 15 | 6 |
| | General | Spiritual | 8 | 28 |
| | | economic | 12 | 22 |
| | Don't know | | 18 | 33 |
| **Coping mechanisms** | Realized | Adapt fishing | 71 | 22 |
| | | Continue | 11 | 33 |
| | | Decrease fishing | 18 | 44 |
| | Hypothetical | Decrease fishing | 65 | 18 |
| | | Continue | 10 | 76 |
| | | Adapt | 25 | 6 |

[×] Questions for which multiple answers per person were possible (total may not add up to 100).

Results are based on interviews with 127 fishers, including 109 men and 18 women.

fishery practices to past changes but would reduce fishing if declining catches continued; women's responses to past changes were heterogeneous, and they would continue fishing if the decline continued. Indeed, in response to the decline in fish abundance, men's responses

were skewed towards adaptation: most (71%) indicated that they had adapted their fishing effort, which included responses related to spending more time at sea, further offshore, or to changing gear. Some (18%) indicated that they had decreased their fishing effort to develop alternative sources of incomes; fewer (11%) reported that they would continue fishing (no changes in fishing effort) (Table 3). Many women had decreased their fishing effort (44%), other continued fishing (33%), and the remaining ones had adapted their fishing practices (22%). In contrast, when asked about what they would do to cope with further decline in catches, most fishermen stated that they would decrease their fishing effort (65%). Alternatively, many suggested that they would adapt (25%), and some would continue fishing (10%). Fisherwomen would generally continue fishing (75%), and to a lesser extent decrease fishing effort (18%) or adapt their practices (6%).

## Relationship between fishers' perceptions, individual- and site-level factors

Overall, the KMO analysis indicated sampling adequacy (S7 Table). The five dimensions recommended for retention under parallel analysis explained 53.48% of the total variance in the data, of which 28.71% was accounted for by the first two axes (S1 Fig). Support for restriction was significantly associated with dimension one (|v.test | = 4.82): however, it was not significantly associated with the remaining dimensions (|v.test statistic| = 1.43, 0.83, 0.86, 0.69).

Dimension one was mostly correlated with coping mechanisms (26.35%), individual characteristics (26.30%), and perceptions related to ecological health (25.32%) (Fig 4A, S8 Table). Variables most positively associated with the first dimension included variables related to social status (gender = women, no membership to fishing associations), individuals who were

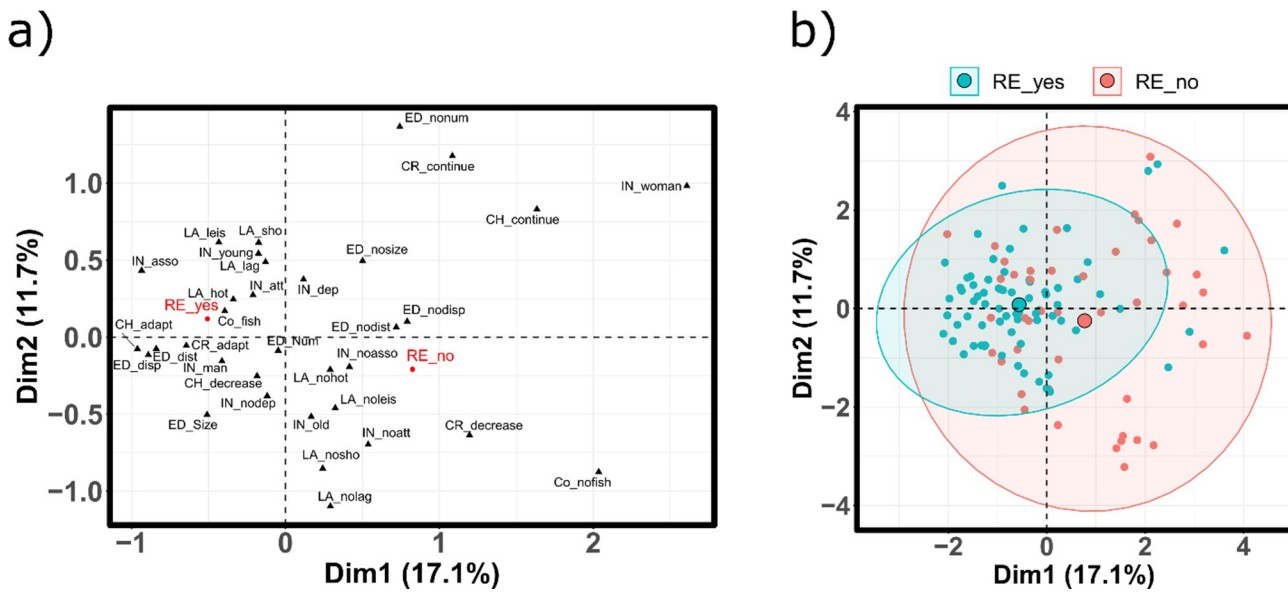

**Fig 4. Visualization of quantitative variables (a) and individuals (b) from a Multiple Factor Analysis (MFA) describing the relationship between support for conservation, individual, and contextal variables for dimension one and two.** On Fig 4A, Individual attributes (IN) include age (young/old), gender (woman/man), association membership (asso/noasso), attachment to fishing (att/noatt), dependence on fisheries (dep/nodep); perceptions related to ecological degradation (ED) include distance to fishing sites (dist/nodist), fish size (size/nosize), fish numbers (num/nonum), and local disapearance (disp/nodisp); perceptions related to underling causes (Co) associated with local fishing (fish/nofish); local assets (LA) include shops (sho/no), hotels (hot/nohot), leisures (leis/noleis), and lagoons (lag/nolag); coping mechanisms include realized (CR) to observed changes (continue/decrease/adapt) and hypothetical (CH) to further changes (continue/decrease/adapt). Support for restrictions (RE) is shown in red (yes/no). On Fig 4B, the small circles are individuals in support (blue) or not in support (red) for fishery restrictions; large symbols indicate mean group values and the elipses show the confidence intervals around group mean points.

likely to have reduced their fishing efforts to cope with changes and responded that they would continue fishing in a hypothetical scenario of further decline, negative response to changes in ecological health (no local disappearance, changes in fishing site distances, or fish sizes) and underlying causal mechanism (not local fishing) (Fig 4A, S9 Table). Support for restriction was negatively associated with this dimension: it was associated with men, perceptions of impaired ecological health attributed to local human agency, and fishers who chose adaptation as a response to current and future changes in the fisheries. While individuals who supported fishery restrictions were clustered on the negative side of the first dimension, the coordinates of those who did not support fishery restrictions were spread across the MFA space (Fig 4B).

The second dimension was most represented by local assets (32.63%) and individual variables (29.49%). Variables positively associated with this dimension included younger individuals, local assets (lagoons, shops, leisure), continued fishing as a reaction to past changes (S10 Table). The third dimension was associated with perceptions related to ecological health (44.90%) and individual characteristics (26.66%). Variables most positively associated with this dimension included younger fishers and the perception that the number of fish had not declined (S11 Table). Dimension four was associated with coping mechanisms (58.20%) and individual characteristics (27.58%); specifically, individuals that responded that they would adapt to further changes in the fishery and those who were dependent on the fisheries were positively associated with this dimension (S12 Table). Alternatively, individuals that did not exclusively depend on the fisheries were associated with decreasing fishing effort as a response to further decline. Finally, dimension five was associated with local assets (42.09%) and individual characteristics (36.24%); specifically, fishers of towns with more hotels were less attached to fishing (S13 Table).

## Discussion

In Madagascar and other parts of the world, the exploitation of marine resources is closely tied to the economy and livelihoods of coastal communities [19, 65, 66]. Small-scale fishers around the world, however, have been reporting declines in their catches [67, 68]. Understanding drivers of local support for fishery management is crucial to implement effective interventions for sustainable fishery [12, 69]. Additionally, fishers' knowledge and perceptions can be an invaluable source of information on fisheries [37, 38]. In agreement with the literature, many interlinked factors were associated with management-oriented attitudes (in our case, the propensity to suggest management restrictions). Factors operating at the individual level, however, were more important than site-level differences. Finally, we show that compiling local perceptions helped understand the current condition of Sainte-Marie's fisheries and inform local fishery management planning by providing a thorough understanding of past socio-ecological changes.

### Long-term socio-ecological dynamics in Sainte-Marie

Our analysis of fishers' perceptions highlights socio-ecological changes as a major force shaping the fishery's temporal trajectory. Indeed, fishers described that the apparition of new markets, combined with limited livelihood options, drastically changed how locals interacted with the fishery. Specifically, more people began to fish, invest in fishing gear, and sell their catches. Due to these changes, fishery landings increased from the 1960s to the 1990s; however, this upward trend shifted in the 1990s. For instance, fishers began to perceive a decrease in fish abundance, catch sizes, and fish sizes; they witnessed the local disappearance of Dugongs (*Dugong dugong*), whose last official sighting in the area dates from 1993 [70], and two Mullidae who used to gather seasonally in large aggregations on Sainte-Marie's shorelines. To

sustain their catches, most fishers adapted their fishing practices; they fished longer, harder, with more efficient gear. Owing to the perception of reduced fish abundance in nearshore areas, many began to fish further offshore. Collectively, the socio-ecological changes described here can lead to the serial depletion of nearshore areas [64] and are known to drive a "race for fish" thought to be an important contributor to overfishing [21, 22, 71].

The consequences of this "race for fish" were indeed perceived by the fishers. For example, they identified an increased number of fishers, followed by the evolution of fishing gear (e.g. less selective, longer nets) as the main cause of the decline. Beyond local fishing, Sainte-Marie's fishers also attributed the decline to overlapping fisheries. In particular, there was a strong perception that sea cucumber fishers, described as sporadic migrants using SCUBA-gear and motorized boats, compete with traditional fishers for marine resources. Although the use of SCUBA to collect marine resources is illegal in Madagascar, similar migrants operate without consequences in many areas [72]. Overall, the socioecological changes described here, as well as their timing, are in agreement with other studies of fishers' perceptions and catch data in Madagascar [21, 22, 39, 71, 73–76]; collectively, these results suggest that socio-ecological changes occurring at both local- and national-scale contribute to the degradation of the fishery by increasing competition for marine resources.

## Relationship between support for management restrictions, individual- and site-level factors

Overall, support for management restrictions was associated with variables implying individual commitment to the fisheries. For example, instead of reducing their fishing effort to develop alternative livelihoods, many fishers who suggested management restrictions had opted to adapt their fishing practices as a response to past changes; they tended to be members of local fishing associations, and shared an understanding of fishery changes and their underlying human causes. Those committed fishers -which were typically men- likely interpreted the current state of the fisheries as a threat to their livelihoods, thus providing an impetus to improve ecological health via fishery restrictions [9, 77, 78]. As such, our results align with other studies demonstrating a relationship between levels of personal bounding to fishery resources, awareness of ecological impairments, and conservation-oriented attitudes [12, 79, 80].

These conservation-oriented attitudes, however, were not associated with conservation-oriented coping mechanisms. Instead, they were associated with fishers who would respond to future changes by further adapting their fishing. This result, while seemingly contradictory, is consistent with studies showing that committed fishers may not be willing to exit fisheries [8, 81, 82]. This suggests that fishers who have adapted their fishing to past changes might displace their fishing effort as a response to future fishery restrictions, thereby coping with changes in a potentially unsustainable way even though they support fishery management restrictions [83].

Similarly, younger fishers were not associated with conservation-oriented coping mechanisms. In contrast to older fishers who were associated with decreased fishing effort, younger fishers continued fishing as a response to past changes. This result is contradictory to other studies where younger fishers were more adaptable and willing to exit the fisheries than their older counterparts [6, 84]. In Sainte-Marie's context, continued fishing might be more of a coping strategy than an adaptive mechanism; younger fishers might therefore opt to reduce fishing if livelihood alternatives become available. Alternatively, younger fishers' willingness to decrease fishing efforts might be influenced by shifting baselines [85]. Indeed, fishers generally believed that fisheries health was not as it once was; however, fishers who perceived no decline in fish abundance tended to be younger. Additionally, older fishers had experienced higher catches associated with lower fishing effort. They recalled childhood experiences where people

fished effortlessly in nearshore areas with traditional gear (e.g. spears, traps), and painted vivid pictures of fishermen catching mullet when the fish aggregated in large numbers on Sainte-Marie's shorelines, before their collapse. As beliefs regarding what healthy fisheries should look like are shaped by fishers' personal experience, younger fishers could unintentionally accept sparser and potentially less diverse fish populations as a baseline, which might influence support for fishery management [86, 87].

Based on previous research, we expected economically-developed towns to be associated with both conservation-oriented attitudes and willingness to decrease fishing, as fishers living in economically-developed towns might have access to more livelihood options [7, 13, 88]. Instead, sites with evidence of economic development were associated with fishers who continued fishing as a reaction to fishery changes; additionally, there was no correlation between economic development and conservation-oriented attitudes. Fishers' willingness to exit fisheries was similarly found to be negatively associated with contextual indicators of socioeconomic development in a broad-scale study across five countries in the Western Indian Ocean [89]. Our result extends these findings by suggesting that similar relationships can occur at finer geographic scales. In our study, it is possible that economic development does not improve fishers' livelihood options; for instance, shops are often family-owned, and the local nightlife may preferentially hire other demographics. Alternatively, access to markets, technology, and in our study the proximity to lagoon (which attract tourists) could also reward fishing in economically developed areas.

## Local fishery management

Fishery management restrictions were often suggested to improve fishery health. While perceptions related to potential restrictions were heterogeneous, fishers were collectively more supportive of area-based restrictions than other forms of management. As in many LMMAs in Madagascar, focusing on small strategic areas for management, including restrictions on less selective gear or minimum harvesting sizes, could help protect marine resources and build support for fishery management in the short-term [11, 27, 28]. However, the current condition of the fishery is tightly interlinked with a shift from subsistence to commercial fishery that has altered the scale of the exploitation of marine resources [90]. Consequently, a successful solution will likely need to involve collaborations between the actors contributing to fishery sustainability, including fishers, NGO, and the fishing industry, to develop innovative solutions and achieve shared long-term goals.

Although developing alternative livelihoods could increase fishery sustainability by reducing fishing pressures, our results suggest that it should be approached cautiously in Sainte-Marie. Indeed, we documented persistent fishing behaviours as a response to past changes, as well as a disconnect between fishers' willingness to decrease fishing and their propensity to suggest management restrictions. Therefore, developing livelihood alternatives would require additional objectives other than increasing fishery sustainability (e.g. risk mitigation, poverty alleviation).

As fishers continue to discuss fishery management rules, good governance will also be key to ensure the success of the initiative. When we conducted this study, fishery management options were often discussed at community meetings. Although these were generally attended by many community members, leadership remained dominated by a select group of powerful individuals that were strongly in support of LMMAs. Acting now to facilitate the effective participation of all fishers, including women, to the elaboration of fishery management plans might be key to ensure the outcomes of fishery management actions [30, 91, 92].

### Caveats and future actions

This study presents data collected when project funding had just started and there was no socio-economic and demographic information for the area. As we did not have any information regarding the number of fishers, sampling efforts were guided by rough estimates, based on consultation with local authorities and communities, of the number of households depending on fisheries in each town. Nonetheless, this study still forms an important assessment of Sainte-Marie's fisheries as it provides a baseline assessment of fishery practices and perceptions before management restrictions were implemented. The timing of this assessment enables future work to provide insights on how fishery management might influence perceptions, and provides a useful baseline for comparison with other small-scale fisheries in Madagascar and elsewhere.

Our data point to gender differences in fishing practices and perceptions that might have consequences for fishery management; however, our conclusions are hampered by the small number of surveyed women. Indeed, despite the broad definition of fishers and fishing adopted during data collection, traditional authorities and respondents generally directed us to men. Our experience highlights that truly engaging with women requires changing how we collect data [45]. Thus, we encourage improving upon the gender-inclusive approach taken here in future research. In particular, we suggest favouring female-led interviewers as women might be more comfortable talking to the same gender [45]. Using multiple data gathering technique could also be beneficial; for instance, conducting interviews in small, women-only focus groups might allow women to share their perspectives [93, 94]. Finally, alternative to traditional participatory research methods (i.e. interview, focus groups), such as the use of narratives blended with photography, could empower women and encourage them to tell their stories [95].

Although we focused on Sainte-Marie, it is important to gain a better understanding of co-existing fisheries. In particular, fishers highlighted their concern about sea cucumber fishers and shrimp trawlers. A better understanding of the impact of those overlapping fisheries, and how to integrate them into management, is critical because local fishers expect that fishery management will help exclude sea cucumber divers, and to a lesser extend shrimp trawlers, from Sainte-Marie's waters. Yet, it should be noted that those broader contextual challenges have proven difficult to address with local fishery co-management initiatives [96–98].

## Conclusion

Local support for fishery management has profound implications for the outcomes and long-term persistence of fishery management initiatives. We found that several interlinked factors were associated with conservation-oriented attitudes in a complex and sometimes counterintuitive way. Overall, conservation-oriented attitudes were associated with fishers with a demonstrated level of commitment to the fishery and extensive ecological knowledge; however, they were not associated with a willingness to reduce fishery use. Additionally, our analysis of fishers' perceptions showed a progressive degradation of the fisheries over the last two decades, that was tightly interlinked with socio-ecological changes in the system. Based on fishers' preferences, area-based management strategy such as LMMA could help build support for conservation. Our results also suggest that steps should be taken to plan creative solutions among the various interest groups involved and build livelihood flexibility to improve fishery sustainability in the long-term.

## Supporting information

**S1 Fig. Scree plot of the multiple factor analysis showing the variance explained by the first ten dimensions.**
(DOCX)

**S1 Table. Main marine resource targets caught in Sainte-Marie, Madagascar, and proportion of the fishers targeting them by fishing method.** In the local fishing vocabulary, common names were often applied to multiple species of fishes. For this reason, we indicate the most representative English fish family, associated with English species name associated with the common Malagasy name.
(DOCX)

**S2 Table. Complete survey administrated to the fishers.**
(DOCX)

**S3 Table. Sample sizes by gender for each of the 16 towns where fishing communities were found in Sainte-Marie, Madagascar.**
(DOCX)

**S4 Table. Generalized additive model selection.** Sp = species, df = degrees of freedom, REML = log-restricted likelihood, Dev. Expl = Deviance explained.
(DOCX)

**S5 Table. Model selection for generalized linear models for temporal changes in fishing distance from the shore.**
(DOCX)

**S6 Table. List of species cited as locally extinct with the number of times they were cited by fishers.**
(DOCX)

**S7 Table. Measure of sampling adequacy (MSA) from KMO test.** KMO > 5 are considered well-sampled; variables with a KMO < could be suspect or have low variance, and should therefore be interpreted with caution. In the case of the number of fishes, fishers almost all indicated a decline. Bartlett's test was significant ($p<0.001$), which indicate that our variables were related and therefore suitable for a factor analysis.
(DOCX)

**S8 Table. Contribution (%) of each group to dimensions one to five of the multiple factor analysis (MFA).**
(DOCX)

**S9 Table. Variables contributing significantly to dimension one and associated categories.**
(DOCX)

**S10 Table. Variables contributing significantly to dimension two and associated categories.**
(DOCX)

**S11 Table. Variables contributing significantly to dimension three and associated categories.**
(DOCX)

**S12 Table. Variables contributing significantly to dimension four and associated categories.**
(DOCX)

**S13 Table. Variables contributing significantly to dimension five and associated categories.**
(DOCX)

**S1 File. R scripts for GAM and GLM.**
(PDF)

**S2 File. R scripts for multiple factor analysis.**
(PDF)

## Acknowledgments

We are infinitely grateful to the fishers of Sainte-Marie for their invaluable support, cooperation, and contributions. We also thank the non-governmental association Cétamada for their technical support. Authors are grateful to the local authorities and traditional chiefs for their invaluable support.

## Author Contributions

**Conceptualization:** Thaïs A. Bernos.

**Formal analysis:** Thaïs A. Bernos.

**Investigation:** Thaïs A. Bernos, Clodio Travouck.

**Methodology:** Thaïs A. Bernos.

**Project administration:** Barbara Mathevon.

**Supervision:** Naly Ramasinoro, Barbara Mathevon.

**Validation:** Clodio Travouck, Naly Ramasinoro, Dylan J. Fraser, Barbara Mathevon.

**Visualization:** Thaïs A. Bernos.

**Writing – original draft:** Thaïs A. Bernos, Dylan J. Fraser.

**Writing – review & editing:** Thaïs A. Bernos, Dylan J. Fraser.

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
