## [Decision Letter · Decision Letter 0]

15 Jul 2021

PONE-D-21-12968

What can be learned from fishers' perceptions for fishery management planning?

PLOS ONE

Dear Dr. Bernos,

Thank you for submitting your manuscript to PLOS ONE. After careful consideration, we feel that it has merit but does not fully meet PLOS ONE’s publication criteria as it currently stands. Therefore, we invite you to submit a revised version of the manuscript that addresses the points raised during the review process.

We look forward to receiving your revised manuscript.

Kind regards,

Hudson Tercio Pinheiro

Academic Editor

PLOS ONE

Journal Requirements:

Additional Editor Comments (if provided):

Dear authors,

I received the feedback of three reviewers and all of them made many comments and suggestions to improve the manuscript. Please take a look in the recommendations and prepare a letter explaining how you addressed each point.

I look forward to see a new version of the manuscript,

Best Regards

Hudson Pinheiro

Reviewers' comments:

Reviewer's Responses to Questions

**Comments to the Author**

1. Is the manuscript technically sound, and do the data support the conclusions?

Reviewer #1: Yes

Reviewer #2: No

Reviewer #3: Partly

2. Has the statistical analysis been performed appropriately and rigorously? 

Reviewer #1: Yes

Reviewer #2: No

Reviewer #3: I Don't Know

3. Have the authors made all data underlying the findings in their manuscript fully available?

Reviewer #1: Yes

Reviewer #2: Yes

Reviewer #3: No

4. Is the manuscript presented in an intelligible fashion and written in standard English?

Reviewer #1: Yes

Reviewer #2: Yes

Reviewer #3: No

5. Review Comments to the Author

Reviewer #1: The manuscript “What can be learned from fishers' perceptions for fishery management planning?” investigated fisher perceptions and empirical knowledge on socioecological aspects of fishing in Madagascar. I read the manuscript with much interest and think is well organized and written, providing important results to subside fisheries management at a local level. However, my overall impression is that the manuscript at current version has a too regional interest, focused from introduction to the discussion in the regional scenario. What are the broader generalizations authors can make with study findings? How can authors situate or not study findings with ‘hot themes’ in ecology? For instance, shifting baselines syndrome, fishing down food webs, policy placebo effects? I think this way authors can make the study interest broader and fit better with the journal scope. Please, see above my specific comments.

Minor comments:

Line 90: “failure” refers to the cancellation/revocation of the MPA implemented?

Linha 127: Oral consent? Please, describe.

Line 154: No text in this section?

Lines 219-220: “By contrast, women’s most frequently 220 perceived causes for the decline were gods’ punishments (44%)”. This response category is not described in the table 3.

Lines 236-237: “most (72%) indicated that they had increased their fishing effort (time at sea, distance from the shore, gear), some (17%) indicated that they had decreased their fishing effort to develop alternative sources of incomes”. The frequencies described in table 3 are 71 and 18. Please double-check it.

Line 259: “We found no significant correlation”. Describe in methods what statistical correlation test was conducted.

Lines 279-281: “suggesting that even younger generations believe that the abundance and diversity of marine resources are not what they once were”. For some species, there is a decreasing trend in catches over the years. Fishers who describe catches earlier perceived higher catches (Fig. 3). Is this an indication of shifting perception over generations?

Figure 1: Inform the location of each village surveyed. Suggestion: add the number for each village name on the map and describe each name in the caption.

Figure 2: Add the title of the y-axis.

Figure 3: Could you add the raw data for each plot? See an example at: https://www.r-bloggers.com/2016/07/plot-some-variables-against-many-others-with-tidyr-and-ggplot2/

Graphical abstract: This is not a graphical abstract, is just figure 1,2 and 3 in the same plate. Please, rethink and present a graphical abstract simple and clearer regarding the study key message.

Reviewer #2: The manuscript PONE-D-21-12968 entitled “What can be learned from fishers' perceptions for fishery management planning?” is a case study from Sainte-Marie, Madagascar. In the study, the authors used interviews with 127 fishers to evaluate the fishery changes. Fishers’ knowledge is a valuable approach which can provide information and help adjust our references of changes in the environment. The manuscript is interesting; however, the study requires major revisions in order to meet the journal's requirements. One important concern is regarding the analysis of changes in the best day’s catch. I think these results could be better exploited when separated by species. I found no justification in the sections of MS explaining why the authors grouped species into families. Both in the introduction and in the discussion lacked references that used the perception of fishers/resource users to promote fisheries management plans. As for the results regarding the perceptions of fishermen and fisherwomen, I see no reason for such gender distinction. Since women were a minority (n=18) in the total number of respondents (n=127), this low sample number does not contribute to a relevant result regarding comparisons related to gender.

An important result that authors not discussed was about the species that disappeared from the catches (lines 212-214). I suggest include a table with these species in the MS. More comments are in the attached PDF. The figures are extremely low quality and unsuitable for publication. After these major revisions, I think that this manuscript is not suitable for publication in a PloS ONE.

Reviewer #3: General comments

Many fisheries management and conservation initiatives fail to prevent overfishing because they do not plan for local engagement and neglect alternatives for short-term losses. Understanding fishers needs and perceptions is of great relevance for fishery management planning because local community support influences the short-term effectiveness and long-term persistence of these management plans. Specifically, where coastal communities strongly depend on marine resources for their livelihood and there is not enough state capacity to enforce fishery rules, as in Sainte-Maire island. This paper contributes to that gap in examining the heterogeneity in fishers’ perception and its contribution to fishery management as it can help uncover opportunities or potential challenges for fishery management, as well as to predict fishers’ potential compliance behavior. Although this study is original and addresses a valid research question, and it is determined by grounding in existing literature, it needs an improvement in the design presentation to become clear and adequate. There is a lack of organization and clear explanation of the methodological approach and data analysis preventing the understanding of the research as a whole, also its replicability. In its actual format, the manuscript is not suitable for publication yet, and major reviews will be needed to fulfill all of the journal criteria for resubmission. In general, I believe that some extra effort has to be made to make the manuscript looks more interesting and organized, so easier to follow and understand. I made few suggestions in the detailed review document.

Specific comments

Tittle: Both the full title and short title were presented differently in the reviewer’s material.

Formatting: There are many formatting problems along all sections of the manuscript, such as reference/citation issues and subheadings organization. Besides, although I am not a native speaker, I believe an English review could be done all over the manuscript to avoid minor grammar errors, spelling, typos, and non-English words, as shown below in these examples:

Line 26: "fshery"

Line 42: omit "locally" or "local" (both are redundant)

Line 43: "rely" instead of “really”

Line 81: “soul-collectors” (is it an English word?)

Line 106: “nation-wide” or nationwide?

Abstract: The abstract is well structured, the authors present the overall aim of the study, its methods, and main results. Finally, they conclude with their findings and the contribution to the field. I have a suggestion about keywords. Keywords ensure that your paper is indexed well by databases and search engines, and thus improve the discoverability of your research. Keywords should contain a list of words that supplement the title’s content, so avoid using terms already present in your title. Therefore, I recommend substituting the keywords “perception” and “Madagascar”.

Tables and figures: In general, including supplementary material, tables and figures should be edited to fix small errors/typos and to improve their legends for a clear explanation of their contents. Tables 2 and 3 could be edited to become shorter. For instance, in Table 2 the main names in the fishing practices (e.g.: mean, gear used, etc.) and socioeconomic characteristics (e.g.: education, status, etc.) may be shown vertically beside their categories. The same could be made in table 3. I do not know where tables and figures will be inserted in the manuscript.

Table 1 – I believe this table is not necessary, the author may describe the survey questions in a small paragraph and add the survey questionnaire in the supplementary material.

Table 2 – what is the difference between “-” and “NA” in this table? Those numbers are percentages? What does “Mean” means in Fishing practices?

Table 3 – Spiritual means divine?

Table S5 – What does “.”, “edf”, and “*” mean?

Figure 1 – It should be call in the line 61.

Figure 2 – What does each bar mean? A village?

Graphic Abstract: The authors could invest extra time to make this infographic more interesting to the readers.

Introduction: The introduction is well written and organized in an easy way to follow and understand. The authors explain the fishing problem and the main research question, focused on the need to consider local support on fishery management and understanding fishers' perception and attitudes. My main concern is regarding the iv) objective (lines 54-55), I do not think the authors have enough data to reconstruct temporal trends in fishery dynamics. These are only three measures of perceived catch for three given years.

Study area and context: The authors did good work describing the study area, and I am really glad to see a rich presentation about the fishing social-ecological system of the region.

Material and methods:

Lines 113 – 120: I believe this paragraph should be relocated to line 154 (“3.2. Ethical considerations”).

Line 124: How the authors chose fishers? How many fishers were interviewed in relation to the total number of fishers in the villages? Did the authors interview all fishers? Did the authors interview the most experienced fishers in the village? What the authors want to say with “randomized and peer-referenced sampling”? It should be better explained.

Line 125: I cannot understand the assumption in the lines 24-25 about fishing techniques, what does it mean?

Line 150-152: I did not understand how the authors did the data validity, lines 150-152 should be better explained.

Line 161: The authors said in line 166 that individual perceptions indexes were only computed for men.

Lines 169 - 173: It was not clear to me how the authors calculated the composite indexes. How was it done the score of an item on the scale of 0-2? Please, provide an example using the research survey questions.

Line 156 - 190: I am not sure the authors applied the correct and transparent methodology/analysis. For example, I did not understand how the GAMM was used to assess perceived catch changes over time with only three measures of perceived catch for three given years. The analysis presented in the S1, regarding fishing efforts, is an important perspective to bring to the manuscript (as shown in the Discussion section). Besides, it could be easy to check first, through multivariate analysis, possible similarities in perception among different groups of fishers or villages.

Results:

Line 193: How many fisherwomen are there in the village? Do the authors believe that the sample size was enough to conclude about differences in perception between fishermen and fisherwomen? Besides, about other assumptions related to gender differences over the manuscript.

Line 203: Why women were rarely members of the local fishers’ association? Are they welcome in those institutions?

Lines 259: The authors did not present a correlation matrix between the composite indexes of perceptions and socioeconomic variables.

Lines 255/263/271: The authors should use the same representation of statistical significance on the manuscript.

Line 268: The authors need to be clear and explain how the model was built and selected. What does “LRT” mean? Species or group of species?

Line 272: Why this figure is in the supplementary material?

Discussion: The authors present their findings and provide useful outcomes that can help fishery management plans in the region. However, the authors need to improve their discussion by comparing their results with other studies and showing the importance of their research to other coastal areas. I acknowledge that considering differences in socioeconomic characteristics of fishers is important to explain the perceptions and responses to fishery management plans. However, the discussion strays towards not really emphasize multiple factors that can act together in defining fishers’ perceptions and attitudes. The authors describing really well the fishing social-ecological system of the studied region, with different environmental and community characteristics among villages. For instance, Southern boroughs have more infrastructure than the others and the eastern boroughs are exposed to the ocean. So, how other factors affect the diversity of responses and perceptions of small-scale fishers? I would recommend accounting for these factors and other mechanisms influencing the perception and attitudes of fishers, including governance aspects.

Conclusions: In this section, I recommend that the authors acknowledge and present the research limitations and gaps.

Finally, in your future work, I think the authors will find the following references to be of use:

• Silas, M.O., Mgeleka, S.S., Polte, P., Sköld, M., Lindborg, R., de la Torre-Castro, M. and Gullström, M., 2020. Adaptive capacity and coping strategies of small-scale coastal fisheries to declining fish catches: Insights from Tanzanian communities. Environmental Science & Policy, 108, pp.67-76.

• Silva, M. R. O., M. G. Pennino, and P. F. M. Lopes. 2020. A social-ecological approach to estimate fisher resilience: a case study from Brazil. Ecology and Society 25(1):23. https://doi.org/10.5751/ES-11361-250123

• Oyanedel, R., Gelcich, S., Milner‐ Gulland, E.J., 2020. Motivations for (non) compliance with conservation rules by small‐ scale resource users. Conservation Letters, e12725.

6. PLOS authors have the option to publish the peer review history of their article (what does this mean?). If published, this will include your full peer review and any attached files.

Reviewer #1: **Yes: **Vinicius J. Giglio

Reviewer #2: No

Reviewer #3: No

---

## [Author Response · Author response to Decision Letter 0]

6 Sep 2021

Dear Dr. Pinheiro, 

Thank you for giving us the opportunity to submit a revised draft of our manuscript “What can be learned from fishers’ perceptions for fishery management planning? Case study insights from Sainte-Marie, Madagascar” [PONE-D-21-12968] for publication in the journal PLOS ONE. We appreciate the time and effort that you and the reviewers dedicated to providing feedback on our manuscript. These comments were very helpful for revising and improving our paper. 

As all reviewers highlighted that the previous manuscript was too locally focused, we entirely rewrote the introduction and discussion. For example, the revised manuscript incorporates discussions of hot themes in ecology, including shifting baseline syndromes and their implications for fishery management. As suggested by one of the reviewers, we also re-analysed the factors influencing support for conservation using a multivariate analysis. This analytical change enabled us to greatly expand the discussion about the linkages between factors operating at the individual-level and the site-level with conservation-oriented attitudes. Finally, another major change is that we present an enhanced description of the methods and include analytical scripts to increase both the transparency and the replicability of the study. As a result of these changes, you will find that the revised manuscript is considerably longer; however, we feel that it is now clearer and of much broader interest to PLOS ONE’s readership. 

Please see below, in bold, for a point-by-point response to the reviewers’ comments and concerns. Those changes are highlighted within the manuscript with tracked changes. In this letter, all page numbers refer to the manuscript file without tracked changes. 

We thank you for your time, 

The authors. 

Reviewer #1: The manuscript “What can be learned from fishers' perceptions for fishery management planning?” investigated fisher perceptions and empirical knowledge on socioecological aspects of fishing in Madagascar. I read the manuscript with much interest and think is well organized and written, providing important results to subside fisheries management at a local level. However, my overall impression is that the manuscript at current version has a too regional interest, focused from introduction to the discussion in the regional scenario. What are the broader generalizations authors can make with study findings? How can authors situate or not study findings with ‘hot themes’ in ecology? For instance, shifting baselines syndrome, fishing down food webs, policy placebo effects? I think this way authors can make the study interest broader and fit better with the journal scope. Please, see above my specific comments.

R: We thank reviewer one for taking the time to provide valuable insights on our manuscript. We agree with them that the previous version of our manuscript was very regionally focused on Sainte-Marie’s fishery. 

Accordingly, we have made major changes to the manuscript, making it of much broader interest to the scientific audience of the journal. The new manuscript focuses on the relationship between support for fishery restrictions and interlinked factors operating at the individual (e.g. perceptions, socioeconomic attributes) and the site-level (infrastructures, natural assets) (see discussion, L497-579). We feel that this theme is of broad interest for ecologists, social scientists, and fishery managers, because lack of support can result in non-compliance to restrictions, thereby influencing the outcomes of fishery management initiatives. Our new manuscript also includes a discussion on shifting baselines, and its potential consequences for management (L552-564), and we allude to the topic of policy placebo (L647-652). We also entirely rewrote the discussion to better situate our results within the broader literature (e.g. L511-513; L517-532; L540-545; L571-579), thereby making our findings more general. 

Minor comments:

Line 90: “failure” refers to the cancellation/revocation of the MPA implemented?

R: The observation is correct. We have changed “failure” to “cancellation of the initiative” (L131).

Line 127: Oral consent? Please, describe.

R: We have replaced “obtained their oral consent” by (L164-165):

“After obtaining [local authorities’] permission to conduct research in the area, we asked them to suggest areas where fishers were likely to be located and to identify suitable survey respondents.” 

Line 154: No text in this section?

R: We apologize as the text had been inserted in the wrong section. We have updated the ethical disclosure section accordingly. 

Lines 219-220: “By contrast, women’s most frequently 220 perceived causes for the decline were gods’ punishments (44%)”. This response category is not described in the table 3.

R: We thank the reviewer for pointing out our inconsistent terminology. To make this sentence consistent with table 3, we replaced “god’s punishment” by “spiritual” (L363). 

Lines 236-237: “most (72%) indicated that they had increased their fishing effort (time at sea, distance from the shore, gear), some (17%) indicated that they had decreased their fishing effort to develop alternative sources of incomes”. The frequencies described in table 3 are 71 and 18. Please double-check it.

R: We have fixed this error (72 was replaced by 71%) (L383).

 “We found no significant correlation”. Describe in methods what statistical correlation test was conducted.

R: We thank the reviewer for pointing this out. As part of our revision process, we have updated the analytical approach used in the analysis of the factors influencing support for management; because of those changes, we have removed the correlation analysis from the manuscript. 

Lines 279-281: “suggesting that even younger generations believe that the abundance and diversity of marine resources are not what they once were”. For some species, there is a decreasing trend in catches over the years. Fishers who describe catches earlier perceived higher catches (Fig. 3). Is this an indication of shifting perception over generations?

R: We agree with the reviewer that elaborating on this point is interesting and have revised the discussion to include consideration on shifting baseline syndrome (L552-564): 

“Alternatively, younger fishers’ willingness to decrease fishing efforts might be influenced by shifting baselines. Indeed, fishers generally believed that fisheries health was not as it once was; perceptions of local extinctions, reduction in fish sizes, and changes in fishing sites, were not associated with age. However, fishers who perceived no decline in fish numbers tended to be younger. Additionally, older fishers had experienced higher catches associated with lower fishing effort. They recalled childhood experiences were people fished effortlessly in nearshore areas with traditional gear (e.g. spears, traps); they painted vivid pictures of fishermen catching mullet when the fish aggregated in large numbers on Sainte-Marie’s shorelines, before their collapse. As beliefs regarding what healthy fisheries should look like are shaped by fishers’ personal experience, younger fishers could be more tolerant of ecological degradation. They could unintentionally accept sparser and potentially less diverse fish populations as a baseline, which might influence support for fishery management.”

Figure 1: Inform the location of each village surveyed. Suggestion: add the number for each village name on the map and describe each name in the caption.

R: We thank the reviewer for their suggestion. We updated figure 1 as suggested: we now show each village as a number on the map, and the names are spelled out in the caption. 

Figure 2: Add the title of the y-axis.

R: As a result of analytical changes, Figure 2 was removed from the manuscript. However, we have ensured that all figures were correctly labelled.

Figure 3: Could you add the raw data for each plot? See an example at: https://www.r-bloggers.com/2016/07/plot-some-variables-against-many-others-with-tidyr-and-ggplot2/

R: We agree and have updated Figure 3 as suggested: it now includes rugs on the margins of the plot as well as points to show the distribution of the data. 

Graphical abstract: This is not a graphical abstract, is just figure 1,2 and 3 in the same plate. Please, rethink and present a graphical abstract simple and clearer regarding the study key message.

R: The new version of our manuscript does not include a graphical abstract. 

Reviewer #2: The manuscript PONE-D-21-12968 entitled “What can be learned from fishers' perceptions for fishery management planning?” is a case study from Sainte-Marie, Madagascar. In the study, the authors used interviews with 127 fishers to evaluate the fishery changes. Fishers’ knowledge is a valuable approach which can provide information and help adjust our references of changes in the environment. The manuscript is interesting; however, the study requires major revisions in order to meet the journal's requirements. One important concern is regarding the analysis of changes in the best day’s catch. I think these results could be better exploited when separated by species. I found no justification in the sections of MS explaining why the authors grouped species into families. 

R: We appreciate the reviewer’s insightful suggestion regarding the analysis of best day’s catch and agree that it would be useful to conduct it at the species-level. However, this was not feasible with our dataset. 

Indeed, when collecting data on catches, we relied on local fishing vocabulary. In Sainte-Marie, the common names used by local fishers to designate fish and marine resources were typically associated with multiple English/Latin species name. We suspect that this is related to their fishing practices; as they generally fish for several species, they may not care too much about subtle differences in their appearance. The analysis was thus performed at the level of the Malagasy species name, which we translated to its most representative family (Table S1) in the previous manuscript. We clarified this point by adding the following sentences to the manuscript (L239-240): 

“The species term corresponded to the common Malagasy name, which often applied to multiple fish species” 

We also improved the caption of table S1 as follow (L1078-1081):

“In the local fishing vocabulary, common names were often applied to multiple species of fishes. For this reason, we indicate the most representative English fish family, associated with English species name associated with the common Malagasy name” 

We also would like to point out that, while addressing the modifications requested by the reviewers, we reanalyzed the temporal trends in catches and found the negative binomial family to produce a better fit. While our conclusions regarding temporal trends in the fisheries remain unchanged, this changes our results slightly; in particular, the interaction between species (=Malagasy common name) and year became unsignificant. Consequently, the results (L418-426, Fig 3) and their discussion (581-632) focus on the fishery/marine resources, rather than the species. Instead of focusing on the ecological health of the area, we feel that our analysis of catch sizes as seen through fishers’ perspectives provides a unique window on the temporal changes in the socio-ecological system as a whole. 

Both in the introduction and in the discussion lacked references that used the perception of fishers/resource users to promote fisheries management plans. 

R: We added several references to other studies using perceptions to promote fisheries management plans (e.g. L44, L89, L490, L513, L537, L562, L594…)

As for the results regarding the perceptions of fishermen and fisherwomen, I see no reason for such gender distinction. Since women were a minority (n=18) in the total number of respondents (n=127), this low sample number does not contribute to a relevant result regarding comparisons related to gender.

R: We agree with the reviewer that women were a minority in our total sample, which limits our conclusions related to gender. Consequently, we now acknowledge small sample sizes for women as a limitation of the study in a paragraph in the discussion (L698-708). However, we believe that gender differences should still be discussed for several reasons. 

From a theoretical standpoint, it is widely acknowledged that women are actively engaged in fisheries globally; however, their participations tend to be vastly overlooked in fisheries research and we still have a limited understanding of their participation in fisheries. One of the reasons why gender specific data remains scarce is because collecting data on fisherwomen is sometimes challenging (cf Kleiber et al. 2014). We believe that the challenges we encountered while looking for women respondents are not unique to our study site and hope that other researchers can improve upon our approach. For this reason, we now provide more details on the gender-inclusive method we used to include women in the data collection (L166-171). We also discuss some of the socioecological barriers influencing the participation of women to our survey (L299-308); finally, we suggest improvements to our approach that could be valuable for other researchers (L701-708).

We also would like to point out that, in our paper, we distinguished fishermen from fisherwoman in two instances; to summarize their characteristics and perceptions (Table 2 and 3), and for the Multiple Factor Analysis. The summary is a qualitative discussion illustrating fishers characteristics and perceptions; in those situations, it is impossible to truly assess what a sufficient sample size might be, as even individual samples might provide valuable information to direct future research (Boddy 2016). Based on our empirical knowledge of the area, there are important gender-based differences in terms of circumstances, involvement, barriers, and benefits that cannot be ignored. The differences in fishing practices and socioeconomic characteristics among gender emerging from this study, as well as a recent socioeconomic survey of the area, only confirm those differences. We therefore added the following statement to justify the separate discussion (L335-337): 

“As differences in the circumstances and involvement of women emerging from this study and a socioeconomic survey of the area (GRET, unpublished data, 2021) could influence their perceptions, we discuss results separately among gender in the following section.”. 

In the Multiple Factor Analysis, gender is included as a variable. Prior to running this analysis, we used a KMO test to check sampling adequacy for each of our variable, including gender; the results of this analysis indicated that genders were adequately sampled for such an analysis, and that results can be interpreted confidently. In the revised manuscript, we include the result of the KMO and associated metrics of sampling adequacy in Table S9 (referred to at L434). 

Finally, and despite being hampered by small sample size of women, we believe that our findings are representative of the area. Indeed, while the number of surveyed women is small compared to men, data from a recent demographic survey of the area (we conducted it over the last year) suggests that our survey encompasses, respectively, 2 and 4% of the men and women. It also confirms socioeconomic differences among genders, including membership to fishing associations, education, etc (now at L295). Finally, such differences among men and women are relatively common in the Western Indian Ocean (Murunga 2021). This is now discussed in the manuscript (L328, L540-545). 

Boddy, C. R. (2016). Sample size for qualitative research. Qualitative Market Research: An International Journal, 19, 1–19.

Kleiber, D. L., & Harris, L. M. (2014). Gender and small-scale fisheries : A case for counting women and beyond. Fish and Fisheries. https://doi.org/10.1111/faf.12075

Murunga, M. (2021). Towards a better understanding of gendered power in small scale fisheries of the Western Indian Ocean. Global Environmental Change, 67, 102242.

An important result that authors not discussed was about the species that disappeared from the catches (lines 212-214). I suggest include a table with these species in the MS. More comments are in the attached PDF. 

R: We thank the reviewer for their suggestion and discuss local disappearance in greater length in the current manuscript. For instance, we now include a table in the supplementary material presenting common species name. Among other, we added the following statements to the discussion (L597-604): 

“In particular, they witnessed the disappearance of Dugongs (Dugong dugong), whose last official sighting in the area dates from 1993. Many also mentioned the disappearance of two Mullidae who used to appear seasonally in large aggregations on Sainte-Marie’s shorelines. Sainte-Marie overlaps with the range of three Mullidae species (Crenimugil crenilabis, Liza vaigiensis, Liza macrolepis) known to form spawning aggregations. Because they are often targeted with unsustainable fishing practices, these spawning aggregations are highly vulnerable to overexploitation and were lost in many locations”.

The figures are extremely low quality and unsuitable for publication. After these major revisions, I think that this manuscript is not suitable for publication in a PloS ONE.

R: We have improved the quality of the figures, which are now suitable for publication. 

L8: Heterogeneous? But the most mentioned cause by fishermen was the increase in fishing effort. I suggest including this information in the abstract. 

R: The reviewer’s observation is correct. We now specify that increased fishing effort is perceived as the main cause in the manuscript (e.g. L353, L617), and the abstract includes the following statement (L17): 

“Accordingly, fishers identified increased fishing effort (number of fishers and gear evolution) as the main cause of fishery changes” 

L26: “fishery” 

R: We have corrected the typo. 

L28: “Remove the comma”

R: We largely reframed the introduction and the discussion. As a result of which, this sentence was removed. 

L39: Furthermore, fishing has impacted the marine environment in several ways; include the negative effects of fishing. 

R: We thank the reviewer for their suggestion. We now refer to ecological degradation in a more general way (L51-52); 

“However, the ecological health of the area is threatened by overexploitation, destructive fishing practices, sediment supplies, climate change, and declining fisheries”

L184: Did you use the family and not the species in the analyses? In Figure 3 you showed the temporal trends in the captures of 13 families. This result could be shown by species, or at least a figure with the main species of each family and the rest of the species in the supplementary material. 

R: See response to this comment above. 

L202: Is there no information on the number of registered fishers in the fishing association? 

R: When this study was conducted, information on the number of registered fishers in the fishing associations was not available. We recognize that this limitation should be mentioned in the paper, so we added a paragraph to the limitation section focused on the topic and steps taken to design sampling (L678-688). 

We should however point out that, over the last year, our team has been collecting socioeconomic data in Sainte-Marie’s villages. Preliminary result from this ongoing work based on extensive consultations with local authorities and communities indicate that, in 2021, nearly 500 fishers were associated with 15 local fishing associations. Importantly, these results also suggest that our sample size represent 2 and 4% of, respectively, fishermen and women. We incorporated this new data in the manuscript as follow (L303); 

“we estimate our sample sizes to represent 2 and 4% of, respectively, Sainte-Marie’s fishermen and women”

L212: Very interesting result, where is the list of species? I suggest including in MS. 

R: As suggested by the reviewer, the list of species is now shown in Table S4. 

L218: In the abstract, you mention that the causes are heterogeneous, but 60% of fishermen and 33% of women mention the increase in effort as a cause. This sounds like the main cause.

R: We now mention increased fishing effort as the main perceived cause of the decline. 

L236: Also one of the main causes reported by fishers.

R: See above. 

L269: Change "species" for "families"

R: This sentence was deleted from the current manuscript as the paragraph was rewritten. 

L274: In general, the discussion seemed poor in terms of what gaps the study can fill. I also noticed a lack of examples from other articles that used the perception of fishers/resource users to promote fisheries management plans. Examples of articles such as Gerhardinger et al. (2009) and Wiber et al. (2012) used the same approach, which would make the MS more interesting within a more global context.

https://doi.org/10.1016/j.ocecoaman.2008.12.007

https://doi.org/10.1007/s10745-011-9450-7

R: We agree with reviewer two that our previous manuscript was too narrowly focused on Sainte-Marie. As a result, we have made significant efforts to broaden the scope of the manuscript. Among others, we extensively rewrote both the introduction and the discussion, and re-analyzed some of the data. We also added many references to the main text, including Gerhardinger et al. 2009 (referred to at L88, L490) and Wiber er al. 2012 (Referred to at L85, L89). 

L287: What are the solutions that the authors suggest to fishers? Could be interesting to provide fishery management strategies such as fishing quotas, fishing closure, and fishing exclusion area.

R: We agree with the reviewer that further elaborating on this point is important. In the current manuscript, we have greatly expanded on the solutions suggested to fishers (see L634-675). 

L301: Adjust references in the journal's norms.

R: We have fixed the formatting inconsistencies. 

L302: Need to reword this sentence. I am not clear what this is trying to say.

R: This sentence was removed from the manuscript. 

L316: Clarify what factors you are referring to. Reduction of effort, fishers age, or fishing far from shore? 

R: This paragraph was removed from the manuscript. 

L321: It is not clear to specify what the negative experience is.

R: This sentence was removed from the manuscript. 

L353: Interesting result, I would like to hear more of this throughout MS. (increased fishing effort)

R: As previously answered, we expanded our discussion of the increased fishing effort (L606-632). 

L355: Indicate which document is this information. Are they in table S1 or Figure S1?

R: As per reviewer three’s suggestion, the analyses of temporal changes in fishing sites’ distance from the shore is now included in the main manuscript (methods: 248-254, results: L418-426) and shown in Figure 3.

L263: Adjust references in the journal's norms.

R: We have fixed the formatting inconsistencies.

L376: I feel that you missed talking about these studies throughout the discussion. Examples of studies that also identified the same decline scenario in Madagascar.

R: We now refer to several studies that identified the same decline scenario in Madagascar (e.g. L591, L594).

Table 3: These values are percentages? Please, indicate in legend or the table.

R: We thank the reviewer for pointing this out. We now indicate in the headers of the table that those values are percentages. 

Table 3: Interesting result (decline in fish sizes), can be better explored. 

R: The decline in fish size is now referred to in the discussion. For example; 

 “Fish abundance was almost unanimously perceived to have been affected; many fishers also observed smaller fish sizes, and the disappearance of species from the area” (L595-597)

 “Fishers also considered the evolution of fishing gear as another one of the main factors influencing catches. For example, they explained that nets had become less size-selective, longer, and more damaging for marine ecosystems” (L620-622)

Table 3: Where is the list of extinct species? I didn’t find this important result. 

R: As suggested by reviewer two, we now include the list of extinct species as a supplementary table. 

I suggest highlighting that increased number of fishers/effort was the main cause of the changes and include this result in the other sections of the manuscript.

R: As answered previously, we now stipulate that increased fishing effort was perceived as the main cause of changes (L618). 

Again the increased effort was perceived by fishers.

R: see answer to this comment above. 

Fig1: Difficult to identify where is study area. I would like to suggest include a small map with the African continent to better situate the reader.

R: We thank the reviewer for their suggestion; Figure 1 now includes a small map of the African continent to help readers locate Sainte-Marie. 

Fig 2: I didn't understand this figure there is a lack of information on the axes, the figures are in poor quality.

R: We apologize for this. As a consequence of analytical changes, Figure 2 was removed from the manuscript and replaced by the visualization of a multivariate analysis. We ensured that all figures were properly labelled and of good quality. 

Fig3: I suggest adding p-value and the silhouette of a representative of the families in the corresponding graphics. For example, in Serranidae add a silhouette of a grouper. Change "species groups" for "families"

R: We thank the reviewer for this good suggestion. As we reran some of the statistical analyses to address other reviewers’ comments, we found that model fit was better with a negative binomial family (previously fitted with the poisson family). As a result of this change, the interaction between species group and time became unsignificant. For this reason, we now only have one plot showing changes in best catch sizes instead of 13 (Figure 2). 

Table S1: Are any of these species at risk of extinction or with fishing restrictions? If so, it can draw attention to governance. 

R: We thank the reviewer and agree that this would be an interesting question to address in future research endeavours. However, our data does not allow us to look at species-specific risks of extinction because the common name used to designate fishes and marine resources typically corresponded to multiple species (see answer to previous comments). As previously explained, analyses are now conducted at the fishery-level rather than the “species” (common name)-level. 

Here it looks like Scarus sp. belongs to the Carangidae. One suggestion is to line up the species column after the family name.

R: Thank you for pointing this out. We reformatted the table S1 by adding shading and aligning the text to the top of each cell. The transition between species families is now much clearer. 

Table S4: I would like to hear more of these assumptions throughout the discussion. 

R: As a response to reviewers’ comments, we largely rewrote the manuscript to highlight the theoretical and empirical framework of our research, as well as its contribution to the literature. As a result of this process, we removed Table S4 from the current manuscript. Instead, the assumption are now discussed in the introduction (L38-47), and explicitly linked to our results in the discussion (e.g. L511, L518, L536, L549, L566…). 

Table S5: Did you use the family and not the species in the analyses? In Figure 3 you showed the temporal trends in the captures of 13 families. This result could be shown by species, or at least a figure with the main species of each family and the rest of the species in the supplementary material. Change "species" for "families".

R: As previously answers, the analysis was conducted at the “species” (common name)-level and are now conducted at the level of the fisheries due to analytical changes. 

Fig S1.1. It is difficult to identify what dots correspond to the years, mainly symbol of Free-diving. The figure is very small and of low quality.

R: We apologize for this error and have updated the Figure (now Figure 3). 

Reviewer #3: Many fisheries management and conservation initiatives fail to prevent overfishing because they do not plan for local engagement and neglect alternatives for short-term losses. Understanding fishers needs and perceptions is of great relevance for fishery management planning because local community support influences the short-term effectiveness and long-term persistence of these management plans. Specifically, where coastal communities strongly depend on marine resources for their livelihood and there is not enough state capacity to enforce fishery rules, as in Sainte-Marie island. This paper contributes to that gap in examining the heterogeneity in fishers’ perception and its contribution to fishery management as it can help uncover opportunities or potential challenges for fishery management, as well as to predict fishers’ potential compliance behavior. Although this study is original and addresses a valid research question, and it is determined by grounding in existing literature, it needs an improvement in the design presentation to become clear and adequate. There is a lack of organization and clear explanation of the methodological approach and data analysis preventing the understanding of the research as a whole, also its replicability. In its actual format, the manuscript is not suitable for publication yet, and major reviews will be needed to fulfill all of the journal criteria for resubmission. In general, I believe that some extra effort has to be made to make the manuscript looks more interesting and organized, so easier to follow and understand. I made few suggestions in the detailed review document.

R: We thank reviewer 3 for their valuable insight on our work. In the current manuscript, we have made several changes to improve the clarity of the manuscript and make it more relevant to a broader scientific audience. 

Among other, we extensively rewrote the methods section of the manuscript to make the research easier to follow, understand, and replicate. For instance, we now provide more information on the analytical approaches used in the manuscript (e.g L230-236; L259-264), including why they are particularly well-suited to our data structure. We also included R scripts (R markdown files) in the supplementary materials (Supplementary Files S1 and S2). We have also extensively rewritten both the introduction and the discussion to discuss themes that will be relevant to a broader audience, including the interlinked factors affecting support for conservation (L497-579) as well as shifting baselines (L552-564). We have addressed each of the reviewer’s comment, as detailed in comment-specific answers below. 

Specific comments

Tittle: Both the full title and short title were presented differently in the reviewer’s material.

R: We apologize and have fixed this error. 

Formatting: There are many formatting problems along all sections of the manuscript, such as reference/citation issues and subheadings organization. Besides, although I am not a native speaker, I believe an English review could be done all over the manuscript to avoid minor grammar errors, spelling, typos, and non-English words, as shown below in these examples (L26, 42, 43, 81, 106), 

R: We thank the reviewer for pointing this out and made sure to correct any potential typos, as well as formatting problems. 

Abstract: The abstract is well structured, the authors present the overall aim of the study, its methods, and main results. Finally, they conclude with their findings and the contribution to the field. I have a suggestion about keywords. Keywords ensure that your paper is indexed well by databases and search engines, and thus improve the discoverability of your research. Keywords should contain a list of words that supplement the title’s content, so avoid using terms already present in your title. Therefore, I recommend substituting the keywords “perception” and “Madagascar”.

R: We thank the reviewer for this suggestion. In the current version of the manuscript, we replaced “perception” and “Madagascar” by the following keywords: “marine protected area” and “livelihood”.

Tables and figures: In general, including supplementary material, tables and figures should be edited to fix small errors/typos and to improve their legends for a clear explanation of their contents. Tables 2 and 3 could be edited to become shorter. For instance, in Table 2 the main names in the fishing practices (e.g.: mean, gear used, etc.) and socioeconomic characteristics (e.g.: education, status, etc.) may be shown vertically beside their categories. The same could be made in table 3. I do not know where tables and figures will be inserted in the manuscript.

R: The reviewer makes a great point. We edited tables and figures to fix typos, ensure that the legends clearly explained their content, and made the suggested changes to Table 2 and 3 (see Table 2 and 3). 

Table 1 – I believe this table is not necessary, the author may describe the survey questions in a small paragraph and add the survey questionnaire in the supplementary material.

R: We agree with the reviewer. We removed Table 1 from the current version of the manuscript. We added a section that described the structure of the interviews, including the questions (L176-200) and the questionnaire is in the supplementary material (Table S2).

Table 2 – what is the difference between “-” and “NA” in this table? Those numbers are percentages? What does “Mean” means in Fishing practices?

R: We thank the reviewer for pointing the lack of clarity of this table. We have updated some of the categories, as they summarized our results better. The column headers now specify that the number are percentages “(%)” (Table 1 and 2). We replaced “means” by “Fishing type” (Table 2). 

Table 3 – Spiritual means divine?

R: We replaced “divine” by “spiritual” in the table and throughout the manuscript. 

Table S5 – What does “.”, “edf”, and “*” mean?

R: We now specify all abbreviations in the table, including edf (estimated degrees of freedom). We used “.” And “*” to indicate p-values, which are now specified. 

Figure 1 – It should be call in the line 61.

R: As suggested, we now call Figure 1 where Sainte-Marie is first mentioned (line 63). 

Figure 2 – What does each bar mean? A village?

R: As a result of analytical changes, the figure 2 referred to in this comment was removed from the manuscript. 

Graphic Abstract: The authors could invest extra time to make this infographic more interesting to the readers.

R: We removed the graphical abstract from our submission. 

Introduction: The introduction is well written and organized in an easy way to follow and understand. The authors explain the fishing problem and the main research question, focused on the need to consider local support on fishery management and understanding fishers' perception and attitudes. My main concern is regarding the iv) objective (lines 54-55), I do not think the authors have enough data to reconstruct temporal trends in fishery dynamics. These are only three measures of perceived catch for three given years.

R: First, we thank the reviewer for their feedback on our writing. As for their concern on our analysis of temporal trends, we apologize as we think it stems from a lack of clarity in the method section of our manuscript. 

First, we would like to emphasize that this analysis was based on 562 observations. Specifically, fishers (n=101) estimated their best catch sizes for up to three species, for up to three time periods each. Those time periods included the year before the study was conducted, the year after they started fishing, and (if possible) the year when a major cyclone hit the island (2009). For example, fishers who had been active in the fisheries since before 2009 provided us with three estimates (including 2017, 2009, and their start date); those who started fishing afterwards provided us with two estimates (start date, and 2017). As a result, we had estimates of best catch size for 34 given years between the mid-1960s and late 2010s. We now provide more information on sample size in the results (L409-L411) as well as the supplementary materials (Table S5); data points are also shown on the figures as both rugs and points (Figure 2).

In our model, we included a random effect because we needed to account for repeated measurements within fishers. The number of fishers is largely above the minimum number of random-effect levels recommended for mixed-model estimation (5 to 6). The model fits well, and the model coefficients (see Table S5) do not provide any indication of overfitting or overparameterization. 

Study area and context: The authors did good work describing the study area, and I am really glad to see a rich presentation about the fishing social-ecological system of the region.

R: We thank the reviewer for this comment! 

Material and methods:

Lines 113 – 120: I believe this paragraph should be relocated to line 154 (“3.2. Ethical considerations”).

R: This observation is correct; we have fixed this error. 

Line 124: How the authors chose fishers? How many fishers were interviewed in relation to the total number of fishers in the villages? Did the authors interview all fishers? Did the authors interview the most experienced fishers in the village? What the authors want to say with “randomized and peer-referenced sampling”? It should be better explained.

R: We thank the reviewer for pointing this out and agree that our sampling design required more information. This information is now provided in the following paragraph (L166-171):

“We asked interviewees to refer us to additional fishers (snowball sampling). Sampling was purposive rather than random as we wanted to ensure sufficient representation of various fishing practices and socioeconomic backgrounds. As part of this effort, we employed a gender-inclusive approach. For instance, when looking for suitable respondents, we defined fishers broadly as people who extract marine resources using various fishing methods for commercial or subsistence purposes. We also stated that we were looking for both women and men.”

When the study was conducted, there was no information on the number of fishers, which we now discuss as a limitation of our study in the manuscript (see L678-688). However, our team conducted extensive socioeconomic survey over the last year. These new data suggest that, overall, we sampled 2 and 4% of, respectively, fishermen and women; we therefore included the following statement (L303): 

“we estimate our sample sizes to represent 2 and 4% of, respectively, Sainte-Marie’s fishermen and women” 

Line 125: I cannot understand the assumption in the lines 24-25 about fishing techniques, what does it mean?

R: This sentence was removed from the manuscript as we rewrote this paragraph to address the comment above. 

Line 150-152: I did not understand how the authors did the data validity, lines 150-152 should be better explained.

R: This sentence was removed from the manuscript. 

Line 161: The authors said in line 166 that individual perceptions indexes were only computed for men.

R: In the current version of the manuscript, we changed the analytical approach related to the factors influencing support for conservations. As a result of this change, we do not compute individual perception indexes in the revised manuscript. 

Lines 169 - 173: It was not clear to me how the authors calculated the composite indexes. How was it done the score of an item on the scale of 0-2? Please, provide an example using the research survey questions.

R: In the current version of the manuscript, we changed the analytical approach related to the factors influencing support for conservations. As a result of this change, we do not compute individual perception indexes in the revised manuscript. 

Line 156 - 190: I am not sure the authors applied the correct and transparent methodology/analysis. For example, I did not understand how the GAMM was used to assess perceived catch changes over time with only three measures of perceived catch for three given years. 

R: As previously mentioned, we apologize for not making our analytical choices and methods more transparent in the previous manuscript. We previously addressed the reviewer’s concern regarding our sample size. We also would like to highlight that we made several additional changes to increase the transparency of our analytical approach. 

Importantly, we provide more information on GAM and explain why they are particularly ideal for our data structure and analytical goals in the following paragraph (L230-236)

 “GAMs were ideally suited to the structure of our data and the nature of our analysis because; 1) their non-parametric smoothing function (hereafter referred to as smoothers) allowed us to model nonlinear temporal trends; 2) they can incorporate both continuous and categorical variables; 3) they can accommodate random effects; and, 4) they estimate the shape of the relationship from the data itself (we did not have to specify any a-priori shape). For these reasons, GAM represented a flexible and powerful approach to model temporal trends in best catches, as well as their nature and timing.” 

Second, we provide more detail on model fitting, package selection, and model selection (L236-246). Third, we added more information on model fit (including number of observations: Table S5) and model selection (Table S6) in the supplementary materials. Last but not least, to ensure that our analyses are fully reproducible, we provide R scripts in a supplementary file (supplementary file S1 referred to at L237). 

Line 156 – 190 (ctd): The analysis presented in the S1, regarding fishing efforts, is an important perspective to bring to the manuscript (as shown in the Discussion section). Besides, it could be easy to check first, through multivariate analysis, possible similarities in perception among different groups of fishers or villages.

R: We agree with the reviewer that analysis S1 was an important perspective. As suggested, we have incorporated the analysis to the main body of the revised manuscripts (Methods: L248-254, Results; L418-426, Fig 3) and discuss the results in the discussion (L606-615).

Second, we thank the reviewer for their suggestion to check for patterns using multivariate analysis. In the current manuscript, we use Multiple Factor Analysis (instead of ANOVAS) to investigate the individual and site-level factors influencing support for conservation (Methods: L256-276, Results; L432-482, Figure 4). As a result of this analytical change, the linkages between different factors and their influence on fisher’s perceptions became much more apparent. It is now discussed at length in the discussion (L497-579) and mentioned in the abstract (L8-12). We feel that this new analytical approach greatly contributed to generalize our findings, thus making our paper interesting to a broader audience. 

Line 193: How many fisherwomen are there in the village? Do the authors believe that the sample size was enough to conclude about differences in perception between fishermen and fisherwomen? Besides, about other assumptions related to gender differences over the manuscript.

R: Based on the result from a recent socioeconomic survey, we estimate that we sampled about 4% of Sainte-Marie’s fisherwomen (vs 2% of the men), which we now specify in the manuscript (L303). We agree with the reviewer that women were a minority in our total sample, which limits our conclusions related to gender. This is now acknowledged as a limitation of our work in the discussion (L698-708). 

We also would like to point out that, in our paper, we distinguished fishermen from fisherwoman in two instances; to summarize their characteristics and perceptions (Table 2 and 3), and for the Multiple Factor Analysis. The summary is a qualitative discussion illustrating fishers characteristics and perceptions; in those situations, it is impossible to truly assess what a sufficient sample size might be, as even individual samples might provide valuable information to direct future research (Boddy 2016). Based on our empirical knowledge of the area, there are important gender-based differences in terms of circumstances, involvement, barriers, and benefits that cannot be ignored. The differences in fishing practices and socioeconomic characteristics among gender emerging from this study, as well as a recent socioeconomic survey of the area, only confirm those differences. We therefore added the following statement to justify the separate discussion (L335-337): 

“As differences in the circumstances and involvement of women emerging from this study and a socioeconomic survey of the area (GRET, unpublished data, 2021) could influence their perceptions, we discuss results separately among gender in the following section.”. 

In the Multiple Factor Analysis, gender is included as a variable. Prior to running this analysis, we used a KMO test to check sampling adequacy for each of our variable, including gender; the results of this analysis indicated that genders were adequately sampled for such an analysis, and that results can be interpreted confidently. In the revised manuscript, we include the result of the KMO and associated metrics of sampling adequacy in Table S9 (referred to at L434). 

From a theoretical standpoint, it is widely acknowledged that women are actively engaged in fisheries globally; however, their participations tend to be vastly overlooked in fisheries research and we still have a limited understanding of their participation in fisheries. One of the reasons why gender specific data remains scarce is because collecting data on fisherwomen is sometimes challenging (cf Kleiber et al. 2014). We believe that the challenges we encountered while looking for women respondents are not unique to our study site and hope that other researchers can improve upon our approach. For this reason, we now provide more details on the gender-inclusive method we used to include women in the data collection (L168-171). We also discuss some of the socioecological barriers influencing the participation of women to our survey (L301-308); finally, we suggest improvements to our approach that could be valuable for other researchers (L701-708).

Kleiber, D. L., & Harris, L. M. (2014). Gender and small-scale fisheries : A case for counting women and beyond. Fish and Fisheries. https://doi.org/10.1111/faf.12075

Line 203: Why women were rarely members of the local fishers’ association? Are they welcome in those institutions?

R: Short answer is; we are not sure. There are several fishing associations on Sainte-Marie (~14); we know that some of them include women, but very rarely and very few of them. To our knowledge, women are not actively excluded from those institutions, but it could stem from the fact that their participation to the fisheries if often downplayed due to the nature of their activities in the fisheries (see L301-308) or to complex gender-power dynamics. We included the following statement to the revised manuscript (L539-545): 

"Alternatively, differences in the propensity to suggest management restrictions could be explained by gendered-power dynamics. Indeed, in the Western Indian Ocean, gender norms are relatively rigid and roles are often socio-culturally prescribed. For instance, in Sainte-Marie, women participate less in decision-making than men, and fishers’ associations are predominantly men-dominated. As in Kenyan and Tanzanian fisheries, we suspect that women’s lower participation in fishery management could be tied to low acquired formal education and a lack of individual self-confidence.”

Lines 259: The authors did not present a correlation matrix between the composite indexes of perceptions and socioeconomic variables.

R: This analysis was removed from the manuscript because of analytical changes. 

Lines 255/263/271: The authors should use the same representation of statistical significance on the manuscript.

R: We now provide the p-value, rounded to two decimals, as a representation of statistical significance throughout the manuscript. 

Line 268: The authors need to be clear and explain how the model was built and selected. What does “LRT” mean? Species or group of species?

R: As highlighted above, we now provide more detail on the model and model selection in the manuscript (L231-239). 

Line 272: Why this figure is in the supplementary material?

R: We assume that the reviewer was talking about the analysis of temporal trends in fishing distance from the shore as there was no figure at line 272. It is now in the main manuscript (Figure 3). 

Discussion: The authors present their findings and provide useful outcomes that can help fishery management plans in the region. However, the authors need to improve their discussion by comparing their results with other studies and showing the importance of their research to other coastal areas. I acknowledge that considering differences in socioeconomic characteristics of fishers is important to explain the perceptions and responses to fishery management plans. However, the discussion strays towards not really emphasize multiple factors that can act together in defining fishers’ perceptions and attitudes. The authors describing really well the fishing social-ecological system of the studied region, with different environmental and community characteristics among villages. For instance, Southern boroughs have more infrastructure than the others and the eastern boroughs are exposed to the ocean. So, how other factors affect the diversity of responses and perceptions of small-scale fishers? I would recommend accounting for these factors and other mechanisms influencing the perception and attitudes of fishers, including governance aspects.

R: We thank the reviewer for their insightful feedback. As a result of analytical changes detailed above (the use of a multivariate analysis), the linkages between different factors and their influence on fisher’s perceptions became much more apparent. It is now discussed at length in the discussion (L497-579).

Additionally, our analysis now also investigates the role of site-level characteristics (local assets) on local perceptions and coping mechanisms, as well as its association with individual attributes. In terms of local assets, we focused on the relative number of hotels, shops, and leisure as they are directly related to alternative livelihood options and economic development (see Table 1). We also investigated the role of proximity to coastal lagoon, as they tend to be sheltered from severe weather events and associated with different fishing practices. We feel that those changes contributed to make our results more generalizable to other sites. 

Conclusions: In this section, I recommend that the authors acknowledge and present the research limitations and gaps.

R: We thank the reviewer for their comment. The revised manuscript included a subsection detailing some of the limitations of our work, including limited sample sizes for women, lack of data regarding number of fishers, and need for additional research on specific topics (677-717). 

Finally, in your future work, I think the authors will find the following references to be of use:

• Silas, M.O., Mgeleka, S.S., Polte, P., Sköld, M., Lindborg, R., de la Torre-Castro, M. and Gullström, M., 2020. Adaptive capacity and coping strategies of small-scale coastal fisheries to declining fish catches: Insights from Tanzanian communities. Environmental Science & Policy, 108, pp.67-76.

• Silva, M. R. O., M. G. Pennino, and P. F. M. Lopes. 2020. A social-ecological approach to estimate fisher resilience: a case study from Brazil. Ecology and Society 25(1):23. https://doi.org/10.5751/ES-11361-250123

• Oyanedel, R., Gelcich, S., Milner‐ Gulland, E.J., 2020. Motivations for (non) compliance with conservation rules by small‐ scale resource users. Conservation Letters, e12725.

R: Those were excellent suggestions, which are now referred to in our manuscript (e.g. L33, L532, L611).

---

## [Decision Letter · Decision Letter 1]

6 Oct 2021

PONE-D-21-12968R1What can be learned from fishers’ perceptions for fishery management planning? Case study insights from Sainte-Marie, MadagascarPLOS ONE

Dear Dr. Thais Bernos,

Thank you for submitting your manuscript to PLOS ONE. After careful consideration, we feel that it has merit but does not fully meet PLOS ONE’s publication criteria as it currently stands. Therefore, we invite you to submit a revised version of the manuscript that addresses the points raised during the review process.

We look forward to receiving your revised manuscript.

Kind regards,

Hudson Tercio Pinheiro

Academic Editor

PLOS ONE

Journal Requirements:

Additional Editor Comments (if provided):

Dear authors,

I sent your manuscript to two reviewers and both agreed that you were able to improve the article and follow their previous comments and suggestions. The reviewers are now suggesting specific comments. Please evaluate these considerations and send us a new version of your manuscript, together with a letter showing how you addressed each comment.

Sincerely,

Hudson Pinheiro

Reviewers' comments:

Reviewer's Responses to Questions

**Comments to the Author**

1. If the authors have adequately addressed your comments raised in a previous round of review and you feel that this manuscript is now acceptable for publication, you may indicate that here to bypass the “Comments to the Author” section, enter your conflict of interest statement in the “Confidential to Editor” section, and submit your "Accept" recommendation.

Reviewer #1: All comments have been addressed

Reviewer #3: All comments have been addressed

2. Is the manuscript technically sound, and do the data support the conclusions?

Reviewer #1: Yes

Reviewer #3: Yes

3. Has the statistical analysis been performed appropriately and rigorously? 

Reviewer #1: Yes

Reviewer #3: Yes

4. Have the authors made all data underlying the findings in their manuscript fully available?

Reviewer #1: Yes

Reviewer #3: Yes

5. Is the manuscript presented in an intelligible fashion and written in standard English?

Reviewer #1: Yes

Reviewer #3: Yes

6. Review Comments to the Author

Reviewer #1: Lines 4-8: “The objectives of this study were two-fold. First, we examined the relationship between conservation-oriented attitudes (e.g. fishers’ propensity to suggest management restrictions), individual characteristics, and site-level characteristics for the small-scale fishers of Sainte-Marie Island, Madagascar. Second, we collated local fishers’ knowledge to understand the historical dynamics of the fishery.”

“We examined the relationship between fishers conservation-oriented attitudes (e.g. propensity to suggest management restrictions) and investigated the historical dynamic of fisheries by assessing fishers knowledge in Madagascar”.

Please, note that the order of aims at the end of the introduction is different from the abstract.

Line 221-226: This is a description of an obvious data organization for most of the studies; I believe this entire paragraph can be excluded.

Fig. 1. Please, add the scale of the geographical coordinate in the axis in the bottom map.

Could the authors reduce the Discussion length? There are 16 paragraphs in this section, it is too long.

Reviewer #3: General comments:

The manuscript “What can be learned from fishers’ perceptions for fishery management planning? Case study insights from Sainte-Marie, Madagascar” has improved a lot. I’m glad to see a better explanation of the method and a broader contextualization of the study findings, as well as considerations about the study's caveats and local fisheries management. However, the new version of the manuscript still needs some revision to become suitable for publication. For instance, the manuscript is considerably longer, I believe that the number of pages should be reduced. Besides, the Results and Discussion sections could be better organized, for instance, the authors could keep the same sequence when presenting objectives and subheadings in the results/discussion sections. Please, see below some suggestions.

Specific comments:

Line 4: Try to write the study’s objectives in the same way it was written throughout the manuscript. In the Introduction section they had written in a clearer way (see lines 64-66, and lines 91-92).

Line 62: It should be the last paragraph of the Introduction. Maybe the author could transfer the current last paragraph (line 84) to line 62 but consider rewrite the sentence in lines 92 – 94 to update the order of objectives.

Line 76: This figure should be in the “Study area and context” section, after the first paragraph. Maybe it could be better to put this section in the Material and methods, before “Survey strategy”.

Line 202: I suggest put this section in the Supplementary material, keeping only lines 212- 218 at the end of the “Survey strategy”.

Line 220: I suggest organizing all sections of the manuscript following the same sequence of objectives. For example, the authors may start with the analysis of the temporal changes, followed by the fisher’s perceptions and then, the relationship between support for conservation and individual/site-level factors.

Line 278: Maybe use the word “classified” (or “categorized”) instead of codified.

Line 294: This table may be presented in the Supplementary material.

Line 297: Avoid discuss results in this section, it is not common to cite other studies here (just present the results). The authors should make assumptions and implications in the Discussion section only. For this comment check these lines: 307-308, 321-322, 346-349, 358-360, 393-395, 421-422, 425-426, and 481-482.

Line 339: Maybe you should keep it simple putting just “Fisher’s perceptions” and adding subheadings for each of the topics (e.g.: Changes, Causes, Solutions, and Coping mechanisms) presenting their specifics results in separated paragraphs.

Line 378: I suggest delete this subheading and keep the structure cited above.

Line 397: I suggest named this subheading “Temporal changes in catches and fishing site distance”.

Line 432: I suggest named this subheading “Fisher’s perceptions and individual and site-level factors”.

Line 456 and 465: Fig. 2a and Fig. 2b?

Results Section: Following my suggestion about the sequence and organization of objectives and results (comment above for line 220). I recommend organize following this way: RESULTS � Fisher’s characteristics � Temporal changes in catches and fishing site distance � Fisher’s perceptions/Changes/Causes/Solutions/Coping mechanisms � Fisher’s perceptions and individual and site-level factors.

DISCUSSION: I suggest reducing the number of pages of this section, the authors should be more concise with their ideas. I also recommend organizing subheadings following this way: Long-term socio-ecological dynamics in Sainte-Marie � Relationship between support for management restrictions, individual, and site-level factors �

REFERENCES: Check for duplicates (e.g.: 27 and 86), also there are too many references, maybe the authors should reduce it.

Supplementary material: The authors do not need to present tables with the models' outputs. I suggest deleting the tables S5 and S7.

7. PLOS authors have the option to publish the peer review history of their article (what does this mean?). If published, this will include your full peer review and any attached files.

Reviewer #1: No

Reviewer #3: No

---

## [Author Response · Author response to Decision Letter 1]

13 Oct 2021

Please note that the lines included below in the reviewers’ comments and authors' responses refer to the original version of our manuscript. 

Reviewer #1: 

Lines 4-8: “The objectives of this study were two-fold. First, we examined the relationship between conservation-oriented attitudes (e.g. fishers’ propensity to suggest management restrictions), individual characteristics, and site-level characteristics for the small-scale fishers of Sainte-Marie Island, Madagascar. Second, we collated local fishers’ knowledge to understand the historical dynamics of the fishery.” and “We examined the relationship between fishers conservation-oriented attitudes (e.g. propensity to suggest management restrictions) and investigated the historical dynamic of fisheries by assessing fishers knowledge in Madagascar”. Please, note that the order of aims at the end of the introduction is different from the abstract.

Authors' response (hereafter: "R"): We thank the reviewer for pointing this out. In the revised version of the manuscript, we made sure that the order of aims at the end of the introduction (l62) and in the abstract (l4) were consistent: 

“First, we collated local fishers’ knowledge to characterize the long-term socio-ecological dynamics of the small-scale fishery of Sainte-Marie Island, in Madagascar. Second, we empirically assessed what individual- and site-level factors might influence support for fishery restrictions.” (L4)

“First, we characterized the long-term socio-ecological dynamics of Sainte-Marie’s fishery; second, we empirically assessed what factors might influence support for fishery restrictions.” (L62) 

Line 221-226: This is a description of an obvious data organization for most of the studies; I believe this entire paragraph can be excluded.

R: We agree with reviewer 1. As suggested, we removed the paragraph from the revised version of the manuscript (L221-226). 

Fig. 1. Please, add the scale of the geographical coordinate in the axis in the bottom map.

R: Unfortunately, we are not sure how to address this comment as the map already includes a scale bar to provide a visual indication of distance on the map. However, to further help readers locate the study area, we added the geographical coordinates of the main town in the main text of the manuscript (L101). We are more than happy to adjust the map as needed but would need more clarification about what the reviewer means. 

Could the authors reduce the Discussion length? There are 16 paragraphs in this section, it is too long.

R: In the revised version of our manuscript, we reduced the length of the discussion/caveats by more than 1/3. Specifically, it was reduced from 2740 to 1808 words (18 to 13 paragraphs). 

Reviewer #3: 

The manuscript “What can be learned from fishers’ perceptions for fishery management planning? Case study insights from Sainte-Marie, Madagascar” has improved a lot. I’m glad to see a better explanation of the method and a broader contextualization of the study findings, as well as considerations about the study's caveats and local fisheries management. 

R: Thank you! 

However, the new version of the manuscript still needs some revision to become suitable for publication. For instance, the manuscript is considerably longer, I believe that the number of pages should be reduced. Besides, the Results and Discussion sections could be better organized, for instance, the authors could keep the same sequence when presenting objectives and subheadings in the results/discussion sections. Please, see below some suggestions.

R: The reviewer is correct in that, as a result of addressing reviewers comments during the previous round of review, the manuscript had become significantly longer; we significantly shortened the revised version of the manuscript. In particular, we reduced the discussion/caveats section of the manuscript by more than 1/3 (from 2740 to 1768 words) and removed about 20% of the references (from 121 to 99). As suggested by reviewer 3, we have also improved the organization of the results and discussion and made sure that objectives and subheadings were presented in the same sequence throughout the manuscript. Please see our answers to specific comments below. 

Specific comments:

Line 4: Try to write the study’s objectives in the same way it was written throughout the manuscript. In the Introduction section they had written in a clearer way (see lines 64-66, and lines 91-92).

R: We thank the reviewer for pointing this out. In the revised version of the manuscript, we rewrote the sentence stating the study objectives to improve clarity (L4). As suggested below, we also reorganized the objectives (L64-66 and 91-92) and ensured that they were consistently ordered in the abstract (L4) and the introduction (L62). 

“First, we collated local fishers’ knowledge to characterize the long-term socio-ecological dynamics of the small-scale fishery of Sainte-Marie Island, in Madagascar. Second, we empirically assessed what individual- and site-level factors might influence support for fishery restrictions.” (L4)

“First, we characterized the long-term socio-ecological dynamics of Sainte-Marie’s fishery; second, we empirically assessed what factors might influence support for fishery restrictions.” (L62) 

Line 62: It should be the last paragraph of the Introduction. Maybe the author could transfer the current last paragraph (line 84) to line 62 but consider rewrite the sentence in lines 92 – 94 to update the order of objectives.

R: We agree. As suggested by reviewer 3, we inserted what was the last paragraph (L84) of our previous manuscript at line 62. As a result of this change, we removed the sentence that was L92-94 and updated the order of objectives (L62) as follow; 

“First, we characterized the long-term socio-ecological dynamics of Sainte-Marie’s fishery; second, we empirically assessed what factors might influence support for fishery restrictions. To do so, we 1) assessed perceptions related to changes in the fisheries, their causes, potential solutions, and fishers’ coping mechanisms and 2) examined whether fishers’ propensity to suggest management restrictions correlated with key factors known to influence support for conservation.”

Line 76: This figure should be in the “Study area and context” section, after the first paragraph. Maybe it could be better to put this section in the Material and methods, before “Survey strategy”.

R: During the previous round of review, another reviewer suggested that Figure 1 be cited and included in the introduction, where Sainte-Marie is first introduced. However, we agree with reviewer 3 that the map is better located in the Material and Method section. For this reason, we decided to move Figure 1 from the introduction (L76) back to the Study area and context section, as suggested here by reviewer 3. 

Line 202: I suggest put this section in the Supplementary material, keeping only lines 212- 218 at the end of the “Survey strategy”.

R: We appreciate the reviewer’s insightful suggestion; however, shortening the ethical considerations section by moving parts of it to the supplementary material is not feasible. Indeed, those edits were requested by the editorial board, who specifically requested that these considerations be included in the manuscript, before the article was moved to editorial review. Nonetheless, we have tried to keep this section as short as possible in the revised paper.

Line 220: I suggest organizing all sections of the manuscript following the same sequence of objectives. For example, the authors may start with the analysis of the temporal changes, followed by the fisher’s perceptions and then, the relationship between support for conservation and individual/site-level factors.

R: We received conflicting advice from both reviewers regarding ways to improve the organization of this section. We decided to make the changes suggested by reviewer 1; specifically, we deleted L221-226, which described a process of data organization common to most studies. To incorporate the suggestion of reviewer 3, we also updated the heading of that section to “Statistical analysis” (L220) instead of “Analysis”. The order in which the statistical analyses are described is consistent with that of the abstract, introduction, results, and discussion (i.e. the analysis of temporal changes precedes that of the relationship between perceptions and individual/site-level factors). 

Line 278: Maybe use the word “classified” (or “categorized”) instead of codified.

R: We replaced the word “codified” by “classified (L278). 

Line 294: This table may be presented in the Supplementary material.

R: We appreciate the reviewer’s helpful suggestion. However, during the previous round of review, we received conflicting advice from another reviewer who suggested that the information contained in the previous version of this table be moved from the supplementary material to the main manuscript. We decided to keep the Table 1 in the main manuscript, because presenting the variables/acronyms in this table also makes it easier to interpret the acronyms figuring in Fig4. 

Line 297: Avoid discuss results in this section, it is not common to cite other studies here (just present the results). The authors should make assumptions and implications in the Discussion section only. For this comment check these lines: 307-308, 321-322, 346-349, 358-360, 393-395, 421-422, 425-426, and 481-482.

R: We removed all of the sentences including citations from the result, as well as those where implications were discussed (e.g. L304-308, 321-322, 346-349, 358-360, 393-395, 421-422, 425-426). 

Line 339: Maybe you should keep it simple putting just “Fisher’s perceptions” and adding subheadings for each of the topics (e.g.: Changes, Causes, Solutions, and Coping mechanisms) presenting their specifics results in separated paragraphs.

R: We thank the reviewer for this comment. As suggested, we changed the title of this section from “overview of the fisheries based on fishers’ perceptions” to “Fishers’ perceptions” (L339); we also added “Fishery Changes” (L340), “Causes” (L351), “Solutions” (L369), and “Coping mechanisms” (L378). 

Line 378: I suggest delete this subheading and keep the structure cited above.

R: As suggested, we deleted the subheading and replaced it by “coping mechanism” (L378). 

Line 397: I suggest named this subheading “Temporal changes in catches and fishing site distance”.

R: We thank the reviewer for this suggestion and replaced the previous subheading by “Temporal changes in catches and fishing site distance” (L397). 

Line 432: I suggest named this subheading “Fisher’s perceptions and individual and site-level factors”.

R: We changed the subheading from “Relationship between support for conservation and various predictors” to “Relationship between fishers’ perceptions, individual-level, and site-level factors” (L432). 

Line 456 and 465: Fig. 2a and Fig. 2b?

R: We thank the reviewer for pointing this out. We replaced Fig. 2a and Fig. 2b by Fig. 3a (L456) and Fig. 3b (L465). 

Results Section: Following my suggestion about the sequence and organization of objectives and results (comment above for line 220). I recommend organize following this way: RESULTS � Fisher’s characteristics � Temporal changes in catches and fishing site distance � Fisher’s perceptions/Changes/Causes/Solutions/Coping mechanisms � Fisher’s perceptions and individual and site-level factors.

R: As suggested, the results (L220) are now organized under the following headers; “Fisher’s characteristics”, “Temporal changes in catch sizes and fishing site distance”, “Fishers’ perceptions” (subheaders; “fishery changes”, “Causes”, “Solutions”, “Coping mechanisms”), and “relationship between fishers’ perceptions, individual- and site-level factors”. 

DISCUSSION: I suggest reducing the number of pages of this section, the authors should be more concise with their ideas. I also recommend organizing subheadings following this way: Long-term socio-ecological dynamics in Sainte-Marie � Relationship between support for management restrictions, individual, and site-level factors �

R: We agree with the reviewers’ comment and reduced the length of the discussion/caveat section in the revised version of the manuscript by more than 1/3. Specifically, it was reduced from 2740 to 1768 words (from 18 to 13 paragraphs). Furthermore, as suggested by reviewer 3, we also reorganized the discussion under the following headers: “Long-term socio-ecological dynamics in Sainte-Marie”, “Relationship between support for management restrictions, individual- and site- level factors”, and “Local fishery management”. 

REFERENCES: Check for duplicates (e.g.: 27 and 86), also there are too many references, maybe the authors should reduce it.

R: We thank the reviewer for noticing the duplicates and ensured that there was no other duplicates in the references. We added a lot of reference to the manuscript to address of the reviewers’ comments in the previous round of review, but we agree that there were too many; we therefore reduced the number of references from 121 to 99. 

Supplementary material: The authors do not need to present tables with the models' outputs. I suggest deleting the tables S5 and S7.

R: As suggested by the reviewer, we removed Table S5 and S7 from the supplementary materials.

---

## [Editor Report · Decision Letter 2]

27 Oct 2021

What can be learned from fishers’ perceptions for fishery management planning? Case study insights from Sainte-Marie, Madagascar

PONE-D-21-12968R2

Dear Dr. Bernos,

We’re pleased to inform you that your manuscript has been judged scientifically suitable for publication and will be formally accepted for publication once it meets all outstanding technical requirements.

Kind regards,

Hudson Tercio Pinheiro

Academic Editor

PLOS ONE

Additional Editor Comments (optional):

Dear authors,

Thank you for addressing all the reviewers' comments and suggestions. The article is now suitable for publication,

Sincerely,

Hudson Pinheiro
---

## [Editor Report · Acceptance letter]

4 Nov 2021

PONE-D-21-12968R2 

What can be learned from fishers’ perceptions for fishery management planning? Case study insights from Sainte-Marie, Madagascar 

Dear Dr. Bernos:

I'm pleased to inform you that your manuscript has been deemed suitable for publication in PLOS ONE. Congratulations! Your manuscript is now with our production department. 

Kind regards, 

on behalf of

Dr. Hudson Tercio Pinheiro 

Academic Editor

PLOS ONE